# Single-cell profiling of lncRNA expression during Ebola virus infection in rhesus macaques

Luisa Santus [1,2,13], Maria Sopena-Rios[1,13], Raquel García-Pérez[1,14], Aaron E. Lin [3,4,5,14], Gordon C. Adams [3,4], Kayla G. Barnes [4,6,7], Katherine J. Siddle [3,4], Shirlee Wohl[3,4,8], Ferran Reverter [9], John L. Rinn [10], Richard S. Bennett[11], Lisa E. Hensley[11] ✉, Pardis C. Sabeti [3,4,5,12] ✉ & Marta Melé [1] ✉

Long non-coding RNAs (lncRNAs) are involved in numerous biological processes and are pivotal mediators of the immune response, yet little is known about their properties at the single-cell level. Here, we generate a multi-tissue bulk RNAseq dataset from Ebola virus (EBOV) infected and not-infected rhesus macaques and identified 3979 novel lncRNAs. To profile lncRNA expression dynamics in immune circulating single-cells during EBOV infection, we design a metric, Upsilon, to estimate cell-type specificity. Our analysis reveals that lncRNAs are expressed in fewer cells than protein-coding genes, but they are not expressed at lower levels nor are they more cell-type specific when expressed in the same number of cells. In addition, we observe that lncRNAs exhibit similar changes in expression patterns to those of protein-coding genes during EBOV infection, and are often co-expressed with known immune regulators. A few lncRNAs change expression specifically upon EBOV entry in the cell. This study sheds light on the differential features of lncRNAs and protein-coding genes and paves the way for future single-cell lncRNA studies.

Long non-coding RNAs (lncRNAs) are transcripts longer than 200 bp that lack protein-coding potential. LncRNAs play important roles in a myriad of processes, such as development[1], evolutionary innovation[2], and disease[3]. LncRNAs often regulate gene expression by acting as signaling molecules[4–6], decoys[7], molecular guides[8], or through scaffolding[9]. Importantly, many lncRNAs are important host immune response regulators[10,11]. Specifically, they regulate the maturation and development of lymphoid and myeloid cells[12], mediate pathogen-induced monocyte and macrophage activation, and the subsequent release of inflammatory factors such as cytokines and chemokines[11,13,14].

Despite lncRNAs sharing similar biogenesis with protein-coding genes[15,16], they are distinguishable by a variety of features, such as

[1]Life Sciences Department, Barcelona Supercomputing Center, Barcelona, Catalonia 08034, Spain. [2]Centre for Genomic Regulation (CRG), The Barcelona Institute for Science and Technology, Barcelona, Spain. [3]FAS Center for Systems Biology, Department of Organismic and Evolutionary Biology, Harvard University, Cambridge, MA 02138, USA. [4]Broad Institute of MIT and Harvard, Cambridge, MA 02142, USA. [5]Harvard Program in Virology, Harvard Medical School, Boston, MA 02115, USA. [6]Department of Immunology and Infectious Diseases, Harvard T.H. Chan School of Public Health, Harvard University, Boston, MA 02115, USA. [7]Liverpool School of Tropical Medicine, Liverpool L3 5QA, UK. [8]The Scripps Research Institute, Department of Immunology and Microbiology, La Jolla, CA, USA. [9]Department of Genetics, Microbiology and Statistics University of Barcelona, Barcelona, Spain. [10]Department of Biochemistry, University of Colorado Boulder, Boulder 80303, USA. [11]Integrated Research Facility, Division of Clinical Research, National Institute of Allergy and Infectious Diseases, National Institutes of Health, Frederick, MD 21702, USA. [12]Howard Hughes Medical Institute, Chevy Chase, MD 20815, USA. [13]These authors contributed equally: Luisa Santus, Maria Sopena-Rios. [14]These authors jointly supervised this work: Raquel García-Pérez, Aaron E. Lin. ✉e-mail: lisa.hensley@nih.gov; pardis@broadinstitute.org; marta.mele.messeguer@gmail.com

lower expression levels[16–18], higher tissue specificity[16,17,19,20], lower splicing efficiency[21,22], and differences in their promoter regulation[23]. However, most of these observations arise from bulk tissue analyses; therefore, whether their expression kinetics are driven by overall low expression levels across many cells or by high expression levels in specific cell populations remains unclear. This lack of knowledge at single-cell resolution hampers our understanding of how lncRNAs function and whether their regulation and response upon infection are intrinsically different from that of protein-coding genes.

EBOV is one of the most lethal pathogens to humans, and it is infamously notorious for its high infectiousness and severe case fatality rates[24,25]. In the past, EBOV caused alarming outbreaks; up to the present day, it represents a major global health threat[26]. Previously, bulk tissue transcriptomic analyses improved our understanding of EBOV's evoked host immune response[27,28]. Now, emerging single-cell RNA-sequencing (scRNA-Seq) technologies are refining our understanding of the systemic immune response mounted upon viral infections[28–30] by allowing the dissection of gene expression dynamics in multiple cell populations simultaneously. More importantly, in the case of organisms infected with a virus, scRNA-Seq can identify and profile infected cells separately from uninfected bystander cells and thus, distinguish the host cellular transcriptional response triggered by viral replication versus the inflammatory cytokine milieu. However, previous studies have focused on the host protein-coding gene response and have ignored the role that non-coding genes such as lncRNAs may play in the host response to EBOV infection. This is mostly due to poor lncRNA annotations in non-human primates, the main species of EBOV research.

In this work, we generate multi-tissue bulk RNAseq data from EBOV-infected and not-infected rhesus macaque tissues to expand the lncRNA annotation in this model organism. We then study circulating immune single-cells infected with EBOV in vivo to address the question of how lncRNAs differentially respond to viral infection at single-cell resolution compared to protein-coding genes. Our results question the long-assumed differences between lncRNA and protein-coding genes and identify lncRNAs involved in the transcriptional response elicited upon EBOV infection.

## Results

### De novo annotation largely expands the rhesus macaque non-coding transcriptome

Bulk and single-cell transcriptomic studies in rhesus macaque have reported widespread host gene expression changes upon EBOV infection[30–32]. However, most lncRNAs have been systematically neglected in such studies due to incomplete annotations, especially in rhesus macaque, where the number of annotated lncRNAs is only 28% of that in humans (Supplementary Fig. 1A). To improve the current lncRNA annotation, we generated short-read RNA-sequencing data from 13 tissues (Fig. 1a) of not infected (16 samples) and EBOV-infected (43 samples) macaques. We further combined this data with publicly available blood RNA-sequencing of not infected (21 samples) and EBOV infected (39 samples) macaques[33], adding up to a total of 119 samples and almost 4 billion reads (Supplementary Data 1). To identify novel lncRNAs, we implemented a computational pipeline that performs de novo transcriptome assembly, extensive quality controls, and non-coding transcript selection based on concordance between three different tools (Fig. 1b, Supplementary Fig. 1B) (see "Methods"). Our approach had high accuracy (82%) and specificity (86%) when predicting Ensembl annotated macaque lncRNAs (Supplementary Fig. 1C). In total, we discovered 3979 novel lncRNA genes (5299 transcripts) (Fig. 1b, c), of which 3191 (80%) were intergenic and 788 (20%) were antisense. Consistent with previous work[34], we identified a human lncRNA ortholog for a relatively low number of lncRNAs (528 lncRNAs (14%)) (Supplementary Fig. 1D). Novel and annotated lncRNA transcripts were shorter, with longer and fewer exons compared to protein-coding genes (Mann–Whitney U test,

all $P$-values < $2.2 \times 10^{-16}$) (Fig. 1d, e). We also observed differences in intron length (Supplementary Fig. 2A). All these observations hold true when we analyze intergenic and antisense lncRNAs separately (Supplementary Fig. 2B–E). In line with previous studies in bulk samples[17,19,20,35], both annotated and novel lncRNAs had lower expression levels (Mann–Whitney U test, all $P$-values < $2.2 \times 10^{-16}$) and were expressed in fewer tissues (two-sided Kolmogorov–Smirnov test, $P$-values < $2.2 \times 10^{-16}$) compared to protein-coding genes (Supplementary Fig. 3A, B).

To further assess the expression profile of lncRNAs, we calculated Tau tissue-specificity scores[36]. Tau is a widely-used metric that measures the level of tissue-specific expression of a gene. It ranges from 0 for housekeeping genes to 1 for tissue-specific genes. As expected[21,22], lncRNAs were more tissue-specific than protein-coding genes (Mann–Whitney U test, $P$-values < $2.2 \times 10^{-16}$) (Fig. 1f). We used Tau to classify genes into tissue-specific (Tau > 0.7), intermediate ($0.3 \le$ Tau $\le 0.7$), and ubiquitous (Tau < 0.3) (Fig. 1g). We found a total of 5203 tissue-specific lncRNAs from which 2429 were novel and 2774 were annotated (Fig. 1g, Supplementary Fig. 3C). Then, for each lncRNA, we identified the tissue in which it presented the highest average expression (see "Methods"). Within such tissues, ubiquitous novel and annotated lncRNAs had similar average expression levels, whereas novel tissue-specific and intermediate lncRNAs were more expressed than annotated lncRNAs (Fig. 1h).

In summary, using de novo bulk sequencing of multi-tissue not infected and EBOV-infected samples, we identified lncRNAs that resemble lncRNA reference annotation and double the current lncRNA rhesus macaque gene annotation.

### LncRNAs are systematically expressed in fewer cells compared to protein-coding genes

Bulk tissue studies have established that lncRNAs are more lowly expressed, more tissue-specific, and often have a more time and context-dependent expression compared to protein-coding genes[16,17,19,20,22]. However, whether this signal arises from lncRNAs being lowly expressed across individual cells or from their expression being restricted to only a few cells remains elusive[37]. To address this, we used single-cell transcriptomics data from macaque's peripheral blood mononuclear cells (PBMCs) from Kotliar et al.[30,38]. After quality control (see "Methods"), we selected 38,067 cells and classified them into four major cell types: monocytes, neutrophils, B cells, and T cells (Fig. 2a, Supplementary Fig. 4A, B). Whereas lncRNAs were slightly less expressed on average than protein-coding genes (Mann–Whitney U test, $P$-value = 0.017) (Fig. 2b), differences in the number of cells in which they were expressed were much larger with lncRNAs being expressed in fewer cells (Mann–Whitney U test, $P$-value < $2.2 \times 10^{-16}$) (Fig. 2c–e). In addition, lncRNAs are consistently expressed in a lower proportion of cells than protein-coding genes when we inspected the different cell types separately (Mann–Whitney U test, all $P$-values < $4 \times 10^{-14}$) (Supplementary Fig. 4C). The proportion of cells expressing a gene and its gene expression levels are tightly correlated (Supplementary Fig. 4D). Thus, we tested whether lncRNA expression levels were lower than those of protein-coding genes when expressed in a comparable number of cells. We found no significant differences in the expression levels of lncRNAs and protein-coding genes when they were matched by the proportion of cells in which they were expressed (one-side Wilcoxon signed-rank test, $P$-value > 0.05) (Fig. 2f). Conversely, lncRNAs were expressed in fewer cells compared to protein-coding genes when controlling for median expression levels (one-side Wilcoxon signed-rank test, $P$-value < $2.2 \times 10^{-16}$) (Fig. 2g). These results indicate that a main distinctive feature of lncRNAs is the low number of cells they are expressed in. We wanted to see if we could reproduce these results in humans, where lncRNA annotation is more complete, and by using an independent platform such as 10X Genomics which has a higher yield than Seq-Well[39]. We used

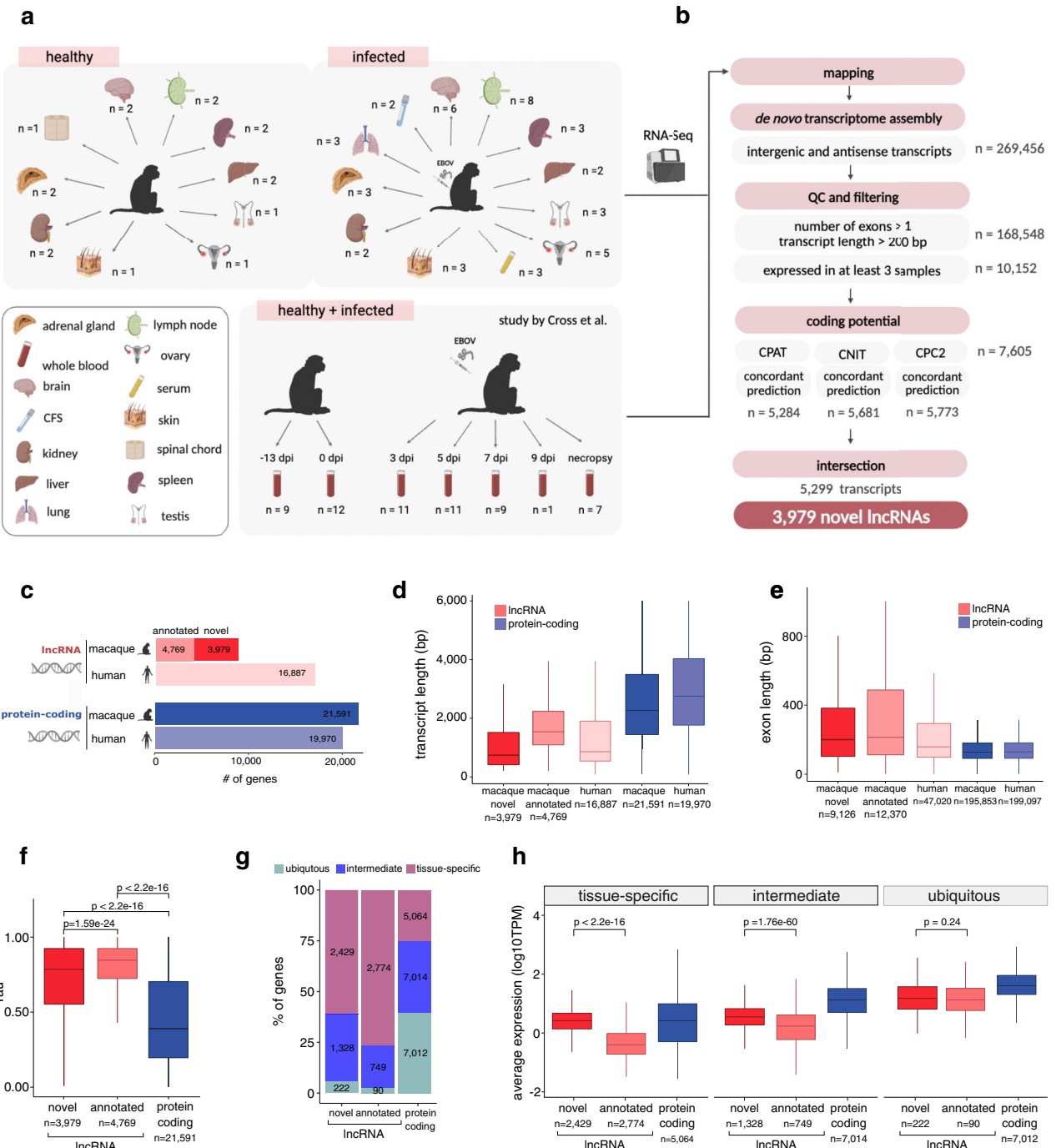

**Fig. 1 | Novel lncRNAs resemble annotated lncRNAs and significantly expand the current macaque lncRNA annotation. a** Samples used for de novo transcriptome assembly. CSF: cerebrospinal fluid. **b** LncRNA discovery pipeline. *N* corresponds to the number of transcripts. **c** Number of novel and annotated lncRNAs and protein-coding genes in the macaque and human annotation (Ensembl release 100). **d** Distribution of transcript length and **e** exon length. **f** Distribution of Tau specificity scores of macaque novel and annotated lncRNA (red) and protein-coding genes (blue). Mann–Whitney U test. **g** Percentage of ubiquitous (Tau < 0.3), intermediate (0.3 ≤ Tau ≤ 0.7), and tissue-specific (Tau > 0.7) lncRNAs and protein-coding genes. Labels indicate the number of genes within each category. **h** Distribution of average expression (log10TPM) in the tissue with the highest expression of tissue-specific, intermediate, and ubiquitous lncRNAs and protein-coding genes. Mann–Whitney U test. *N* corresponds to the sample size of each category. All boxplots display the median and the first and third quartiles of the data. The whiskers extend to the highest and lowest values within 1.5 times the interquartile range (IQR) of the data.

publicly available single-cell RNA-sequencing data from healthy human PBMCs generated with 10X Genomics[40] and replicated our findings (Supplementary Fig. 4E–G). Thus, our observations are consistent regardless of single-cell technology, species, gene annotation or infection status. Overall, our results indicate that in circulating immune cells, the lower expression levels of lncRNA previously reported in bulk studies may be driven by lncRNA being expressed in fewer cells compared to protein-coding genes rather than having less expression across individual cells.

## Upsilon, a metric to measure cell-type specificity in single-cell expression data

Tau is a metric routinely used to measure tissue specificity[36]. However, to our knowledge, no metric to estimate cell-type specificity has been

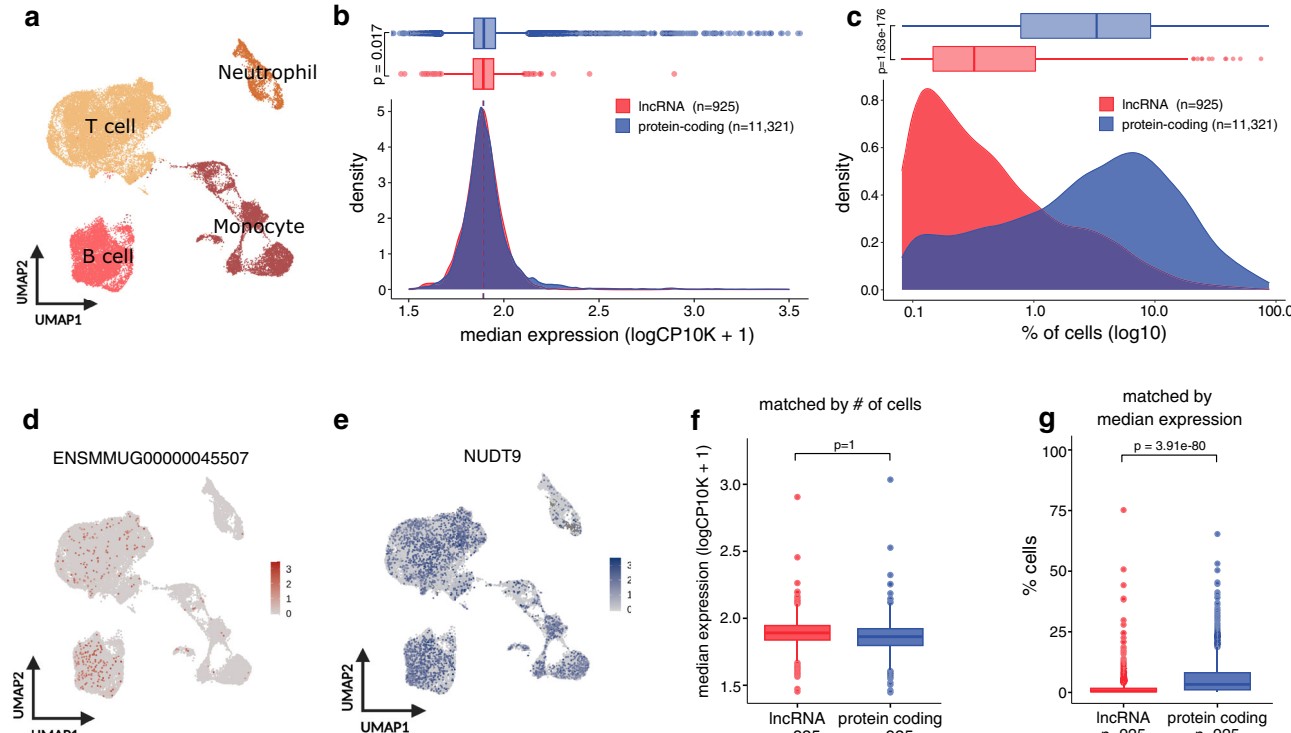

**Fig. 2 | Expression patterns of lncRNAs and protein-coding genes at single-cell resolution. a** UMAP embedding of 38,067 cells. Cell types are indicated by the different colors. **b** Distribution of median gene expression levels (log(CP10K + 1)) and **c** percentage of cells (log10) in which lncRNA (red) and protein-coding (blue) genes are expressed. Mann–Whitney U test. **d** UMAP embedding showing the expression levels of a lncRNA (*ENSMMUG00000045507*) and **e** a protein-coding gene (*NUDT9*), with the same median expression level but expressed in a different number of cells. **f** Distribution of median expression levels of lncRNA (red) and protein-coding genes (blue) when matched by the percentage of cells in which they were expressed. **g** Percentage of cells in which lncRNA (red) and protein-coding (blue) genes were expressed when matched by median expression levels. One-side Wilcoxon signed-rank test. All boxplots display the median and the first and third quartiles (the 25th and 75th percentiles) of the data. The whiskers extend to the highest and lowest values within 1.5 times the interquartile range (IQR) of the data.

established. To address this issue, we designed a metric that estimates cell-type specificity based on single-cell data. We named it Upsilon, which is the next letter in the Greek alphabet after Tau. Whereas Tau relies mostly on differences in expression levels[41], Upsilon relies on the proportion of cells expressing a gene (see "Methods"). Similarly to Tau, Upsilon scores range from 0, for ubiquitous genes to 1, for cell-type specific genes.

In order to evaluate the ability of both metrics to estimate cell-type specificity, we repurposed the Tau calculation. Instead of using the mean expression levels per tissue[41], we used the mean expression levels per cell type (see "Methods"). Both Tau and Upsilon could accurately classify genes as ubiquitous, intermediate, or cell-type specific in simulated scenarios although Upsilon was better at classifying different degrees of intermediate cell-type specificity (Supplementary Fig. 5). In addition, we selected a set of housekeeping and marker genes (see "Methods") and compared their Tau and Upsilon scores. While both metrics assigned low values to housekeeping genes, our metric better identified marker genes as tissue-specific (Fig. 3a).

We then computed Upsilon scores to characterize the cell-type specificity of lncRNAs. Both novel and annotated lncRNA showed similar values (Mann–Whitney U test, P-value > 0.05) (Supplementary Fig. 6A). Using Upsilon, we classified lncRNAs as cell-type specific (Upsilon > 0.7), intermediate (0.3 ≤ Upsilon ≤ 0.7), and ubiquitous (Upsilon < 0.3) (Fig. 3b). We identified 153 cell-type specific lncRNAs, of which 67 (44%) were annotated and 86 (56%) were novel (Fig. 3b, Supplementary Fig. 6B). Previously reported disease biomarkers, such as *MIAT*[42] and *DIO3OS*[43], were among the set of cell-type specific lncRNAs highlighting the utility of our novel metric in identifying candidate genes for diseases. Also, we found that cell-type specific lncRNAs have slightly shorter transcript lengths and slightly fewer and

shorter exons as compared to ubiquitous genes (Mann–Whitney U test, all P-values < $3.5 \times 10^{-3}$) (Supplementary Fig. 6C–E).

Tissue-specific expression of protein-coding genes mainly occurs due to restricted expression at specific cell types[44]. We sought to identify whether this held true for lncRNAs as well. We selected genes that were expressed in both whole blood bulk RNA-seq data and PBMCs single-cell RNA-seq data (see "Methods") (Supplementary Fig. 6F) and compared Tau scores computed in bulk with Upsilon scores computed in single-cell. Tissue specificity was significantly correlated with cell-type specificity both in lncRNAs (Spearman $\rho = 0.31$, P-value < $2.2 \times 10^{-16}$) (Fig. 3c) and in protein-coding genes (Spearman $\rho = 0.46$, P-value < $2.2 \times 10^{-16}$) (Supplementary Fig. 6G) indicating that similar to protein-coding genes, tissue-specific lncRNAs are more likely expressed in particular cell types.

In summary, we developed a metric called Upsilon, which uses single-cell data, to identify and characterize cell-type specific lncRNAs, including known disease biomarkers, demonstrating its potential to pinpoint candidate disease-associated genes.

## The higher specificity of lncRNAs can be attributed to their expression in fewer cells

LncRNAs are known to be more tissue-specific than protein-coding genes[21,22]. We thus wondered whether lncRNA's higher tissue specificity was due to lncRNAs being expressed in fewer cells or to lncRNAs being more cell-type specific. To address this, we compared cell-type specificity values between lncRNAs and protein-coding genes and found that lncRNAs were more cell-type specific (Mann–Whitney U test, P-value < $2 \times 10^{-10}$) (Fig. 3d) and could separate cell types in a UMAP visualization (Supplementary Fig. 7A). However, when matched by the number of cells in which they were expressed, protein-coding

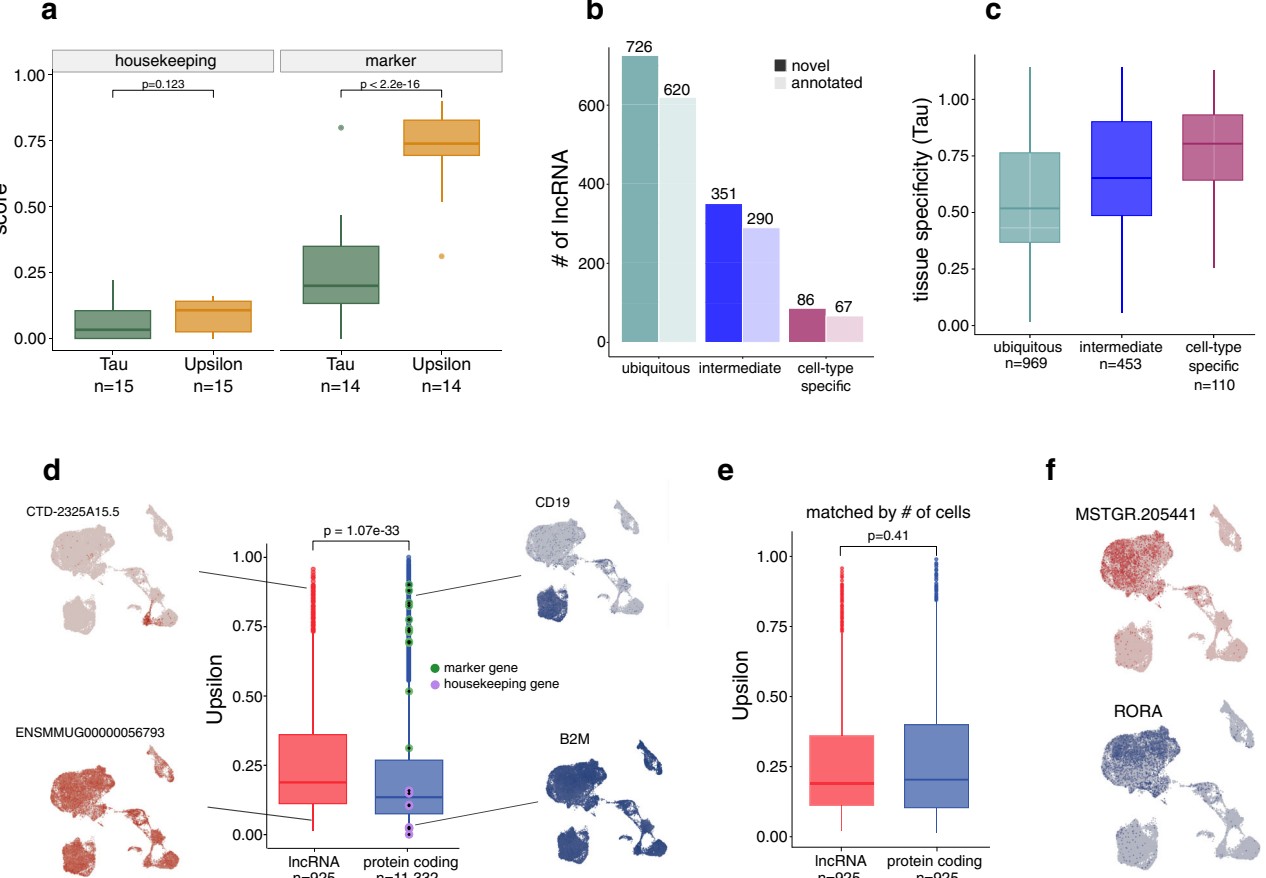

**Fig. 3 | Identification of cell-type specific lncRNAs. a** Distribution of Tau (green) and Upsilon (yellow) cell-type specificity scores for housekeeping (left) and maker (right) genes. Wilcoxon signed-rank test. **b** Bar plot showing the number of ubiquitous (Upsilon < 0.3), intermediate (0.3 ≤ Upsilon ≤ 0.7), and cell-type specific lncRNAs (Upsilon > 0.7). **c** Distribution of tissue-specificity Tau scores of ubiquitous, intermediate, and specific lncRNAs. **d** Distribution of cell-type specificity scores of lncRNA (red) and protein-coding (blue) genes. Cell-type marker genes are highlighted in green, housekeeping genes in purple. UMAP embeddings of cell-type specific and ubiquitously expressed lncRNA (red) and protein-coding (blue) genes

are shown as examples. Mann–Whitney U test. **e** Distribution of Upsilon cell-type specificity scores of lncRNA and protein-coding genes when matched by the percentage of cells in which they were expressed. Wilcoxon signed-rank test. **f** UMAP embedding shows the expression pattern of the cell-type-specific lncRNA *MSTRG.205441* (Upsilon = 0.9) (top) and the protein-coding gene RORA (Upsilon = 0.89) (bottom) which were matched by the percentage of cells in which they are expressed. All boxplots display the median and the first and third quartiles (the 25th and 75th percentiles) of the data. The whiskers extend to the highest and lowest values within 1.5 times the interquartile range (IQR) of the data.

and lncRNA had comparable cell-type specificity scores (Wilcoxon signed-rank test, *P*-value > 0.05) (Fig. 3e, f). On the contrary, when lncRNA and protein-coding genes were matched by their cell-type specificity, lncRNAs were expressed in fewer cells (Supplementary Fig. 7B). To assess whether these observations were independent of species, completeness of lncRNA annotation, infection status, or sequencing platform, we analyzed healthy human PBMC single-cell data. With this dataset, we also observe that lncRNAs are as cell-type specific as protein-coding genes when expressed in the same number of cells (Supplementary Fig. 7C, D).

Overall our observations indicate that the long-assumed higher tissue specificity of lncRNAs derived from bulk studies might be the result of their expression in fewer cells rather than overall higher cell-type specificity.

## LncRNAs are dynamically regulated upon EBOV infection
LncRNAs play crucial roles in the host response to viral infections[45–48]. However, previous studies mostly relied on bulk tissue data, which hinders the detection of expression differences at the cellular level. To investigate the cell-type-specific dynamics of lncRNAs upon immune stimulation, we use single-cell data from in vivo EBOV-infected macaque PBMCs[30]. We sought to identify lncRNAs with immune regulatory roles during viral infections in specific cell types. We performed a

differential gene expression analysis separately in each cell type (monocytes, T, and B cells), comparing each stage of the infection (early, middle, late) to the baseline (see "Methods").

We detected 186 differentially expressed (DE) lncRNAs in at least one cell type (Benjamini–Hochberg's correction, false discovery rate (FDR) < 0.05, fold change >10%) (Fig. 4a–c, Supplementary Fig. 8A–D) (Supplementary Data 2), the majority of which (124 lncRNA, 66%) were novel, underscoring the importance of refining the annotation of lncRNAs in model organisms such as rhesus macaque. The largest number of DE lncRNAs were found in monocytes (142 lncRNAs) (Fig. 4c, Supplementary Fig. 8A–D), consistent with monocytes being the main EBOV target[49,50] as well as the most abundant cell type in our dataset. We then used our cell-type specificity metric, Upsilon, to investigate the cell-type specificity of DE genes. We found that most DE genes were not cell-type specific (Fig. 4d). Of all DE lncRNAs, 34 had a human ortholog, and, 28 of those have been previously reported to change expression during immune response in humans[51] (Supplementary Fig. 8E). Consistent with previous studies of immune response upon infection, *SNHG6* and LINC00861 were upregulated[52–54]. Interestingly, the most transcriptionally repressed lncRNA was the nuclear-enriched abundant transcript 1 (*NEAT1*) (Fig. 4c, e). *NEAT1* is a well-studied lncRNA known to play important anti-viral roles[55,56]. In most studies, however, *NEAT1* is upregulated upon viral infection[57] and

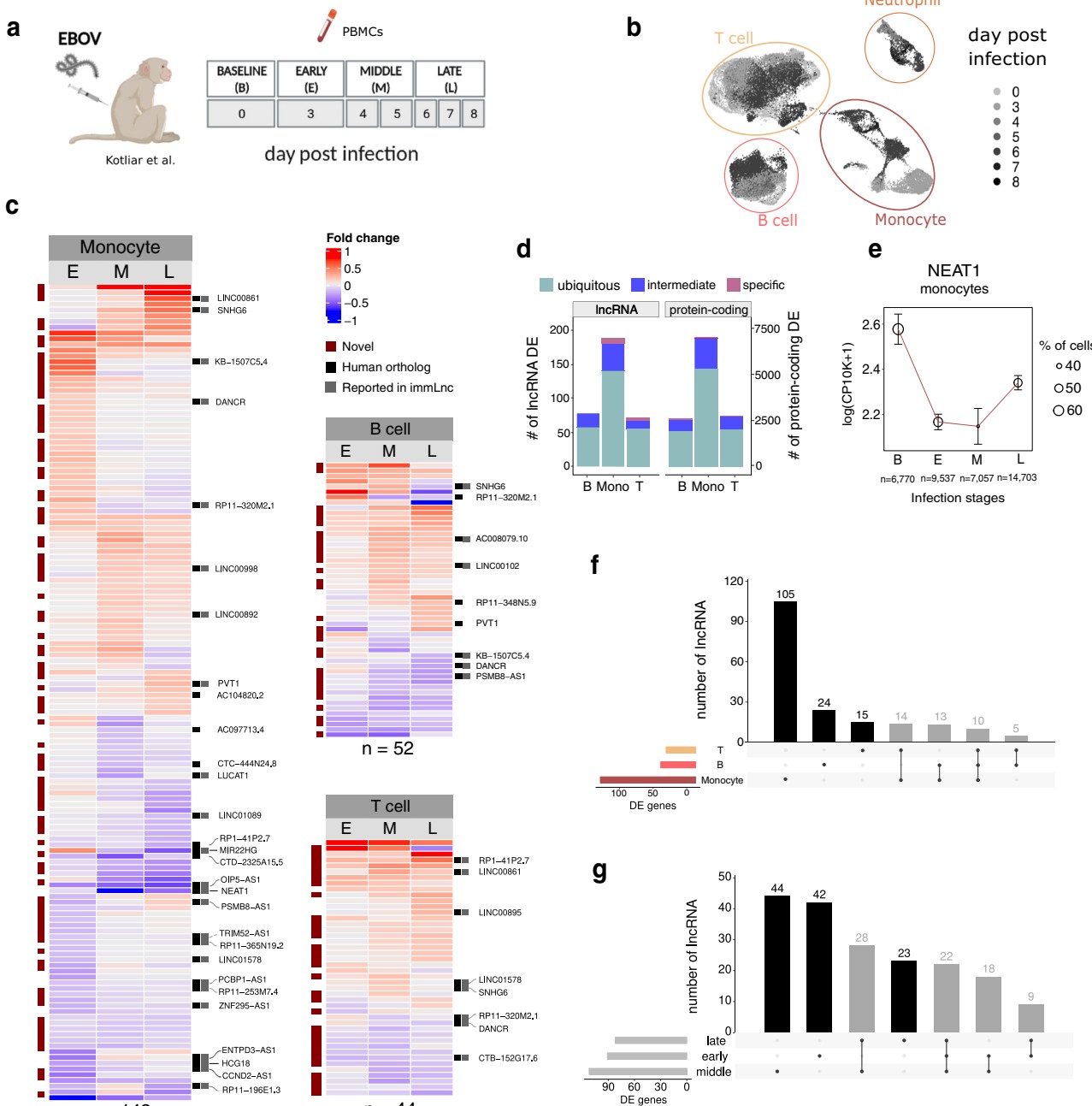

**Fig. 4 | LncRNA expression changes upon EBOV infection are cell-type specific.**
**a** Schematic overview of the in vivo experiment design. **b** UMAP embedding of
38,067 cells from the in vivo dataset, colored by day post-infection (DPI).
**c** Heatmaps display lncRNAs DE in monocytes, T cells, and B cells in at least one
infection stage—early (E), middle (M), or late (L)—as compared to baseline (**b**). Cells
are colored according to the fold changes (log2) in expression values between
baseline and the corresponding infection stage. Only lncRNAs with a human
ortholog have the name displayed. The numbers of DE lncRNAs in each cell type are
depicted at the bottom of the heatmap. **d** Number of DE lncRNAs (left) and protein-
coding genes (right) ubiquitously expressed (Upsilon <0.3), with intermediate cell-
type specificity score (0.3 ≤ Upsilon ≤ 0.7) or cell-type specific (Upsilon > 0.7) in B
cells, Monocytes and T cells. **e** NEAT1 expression pattern at different stages of
infection in monocytes. N corresponds to the number of cells in each reported
infection stage. Dots' sizes represent the percentage of cells in which the gene was
expressed. Dots' centers represent the mean. Error bars indicate the 95% con-
fidence interval around the mean, calculated using the standard error of the mean
(SEM). **f** Upset plots showing the overlap of DE lncRNAs across cell types (**g**) and
infection stages.

downregulation has only been described in dengue and Crimean
Congo hemorrhagic fever[58,59]. Our results suggest that *NEAT1* deple-
tion may be specific to severe hemorrhagic fevers and in the case of
EBOV at least, downregulation occurs specifically in monocytes.

We then wanted to compare the expression dynamics of lncRNAs
to that of protein-coding genes upon immune stimulation. Most
lncRNAs (144 lncRNAs, ~78%) were DE in exclusively one cell type
(Fig. 4f) which was a significantly larger proportion than the one

observed for protein-coding genes (Fisher's exact test, OR = 2.06,
P-value = $1.945 \times 10^{-5}$) (see "Methods"). However, when matched by
the number of cells in which they were expressed, the two gene classes
had comparable proportions of cell-type specific DE genes (Fisher's
exact test; OR = 0.90, P-value = 0.69). Similarly, the majority of
lncRNAs (109 lncRNAs, ~60%) were DE in only one stage of the infection
(Fig. 4g) which is a significantly larger proportion than that of protein-
coding genes (Fisher's exact test, OR = 1.49, P-value = $8.83 \times 10^{-3}$)

(see "Methods"). This difference disappeared when comparing lncRNA and protein-coding genes matched by the number of cells in which they were expressed (Fisher's exact test; OR = 1.03, P-value = 0.57). Overall, our results indicate that upon EBOV immune stimulation the transcriptional response of lncRNAs is stage and cell-type specific similar to that of protein-coding genes.

### Functional characterization of lncRNAs differentially expressed upon EBOV infection

Although we detected many lncRNAs that change their expression upon EBOV infection, most of them remain functionally uncharacterized. Some lncRNAs are known to exert their modulatory role in cis[60]. To identify possible cis-regulatory lncRNAs, we first identified 327 lncRNA protein-coding gene pairs that were both DE in the same cell type and in close physical proximity (<1 Mbp). DE lncRNA and protein-coding genes were not significantly co-located more often than expected by chance (Fisher's exact test, OR = 0.91, P-value > 0.05) (see "Methods"). However, we found 41 gene pairs that were co-located and co-expressed at cell-type resolution (Spearman correlation test, P-value < 0.05, Supplementary Fig. 9).

To explore further the pathways and putative functions of our DE lncRNAs, we built a cell-type-specific co-expression network in monocytes using both lncRNA and protein-coding genes (see "Methods"). The network had 8 modules with an average of 7 lncRNAs and 15 protein-coding genes (Fig. 5a). Three modules displayed significant functional enrichments, primarily related to immune stimulation (Supplementary Fig. 10A–C, Supplementary Data 3). One of these modules contained several interferon-stimulated genes (ISGs), including *MX1*, *IFIT2*, and *ISG15*, and was enriched in genes that increased expression at early and mid stages of infection[30] (Fig. 5a). Interestingly, we identified a lncRNA, *ENSMMUG00000064224*, directly connected to ISGs, that exhibited a similar expression profile as ISG with an upregulation in all three cell types at early infection (Supplementary Fig. 10D, E). We also found one module with a remarkable number of enriched terms related to cell proliferation and migration. Most of the genes in this module were downregulated with the strongest expression changes at the late stages of infection (Fig. 5a), suggesting a late host response to prevent EBOV replication[61,62]. Although the remaining five modules did not have significant enrichments, all of them included between 1 and 8 central regulators or downstream effectors of the innate immune response[63] (Supplementary Data 3).

Previous work based on PBMCs infected with EBOV ex vivo showed that EBOV hijacks infected cells' defenses by downregulating anti-viral genes and upregulating pro-viral genes[30]. Using an ex vivo experimental setup allows for higher viral exposure to EBOV and consequently a higher number of infected cells with higher viral loads compared with the same cell type bystander cells. We sought to investigate if lncRNAs were up or downregulated upon viral cellular entry and proliferation compared to bystander cells. To do this, we identified lncRNAs whose expression significantly correlated with viral load in EBOV-infected monocytes ex vivo (Fig. 5b, Supplementary Fig. 11A–E). We identified 16 lncRNAs significantly correlated with viral load (Spearman correlation test, P-value < 0.05) (Supplementary Data 4), the majority of which (12) were positively correlated (Fig. 5c). Importantly, *ENSMMUG00000058644* and *MSTRG.15458*, which had the strongest correlations, were also significantly correlated at nominal P-values in the in vivo dataset (Spearman $\rho = 0.10$, P-value = 0.03 and Spearman $\rho = -0.12$, P-value = 0.01, respectively), suggesting that the in vivo dataset might not have enough infected cells, and thus power, to identify significant correlations. In line with this, lncRNAs correlated with viral load were expressed in significantly fewer cells in vivo compared to ex vivo (Mann–Whitney U test, P-value < 2 × $10^{-10}$). 10 out of the 16 identified lncRNAs were not detected as DE with EBOV infection in monocytes in vivo (Fig. 4c, Supplementary Data 2

and 4), suggesting that most of these lncRNAs change their expression exclusively in infected cells. This observation highlights the power of the single-cell analysis to discern between expression changes in bystanders and infected cells. Interestingly, the remaining five lncRNAs were DE upon infection in monocytes in the in vivo dataset but in opposite directions: two lncRNAs were upregulated during EBOV infection in the general in vivo monocyte population but were negatively correlated with the viral load in ex vivo infected cells; three lncRNAs were downregulated during EBOV infection in the general in vivo monocyte population but increased their expression with viral load in ex vivo infected monocytes (Fig. 5d–g, Supplementary Fig. 12A–F).

Overall, our functional analyses revealed that lncRNAs whose expression varies upon EBOV infection are involved in the same pathways as DE protein-coding genes, suggesting that these lncRNAs might be important immune regulators. In addition, our ex vivo results indicate that EBOV entry in the cell can alter the expression of lncRNA exclusively in infected cells and that in some cases, the expression changes differ between infected and bystander cells. This would be consistent with previous studies that reported that EBOV hijacks particular pathways in infected cells to promote viral entry and replication[30].

## Discussion

Long non-coding RNAs play critical roles in immune regulation[10,11]. However, studies that require working with non-human animal models, such as Ebola virus infection, are constrained by an incomplete lncRNAs' annotation. To address this issue, we generated a multi-tissue bulk RNA sequencing dataset from both EBOV-infected and uninfected samples and annotated nearly 4000 novel lncRNAs. This effort resulted in nearly doubling the current annotation of lncRNA in rhesus macaque. Importantly, we found that 66% of all lncRNAs changing expression upon EBOV infection in single cells were novel. These findings underscore the importance of expanding current non-coding transcriptome annotations with datasets that sample different physiological conditions, especially in model species widely used in biomedical research[64]. Future work using emerging long-read sequencing technologies[65] will further improve the discovery and annotation of lncRNAs in model species in the context of infection.

LncRNAs are generally assumed to be more lowly expressed and more tissue-specific than protein-coding genes[16]. These observations arise from bulk studies that measure average expression levels across cell populations. Single-cell data allows both detecting gene expression levels in individual cells and determining how many cells in a given population express a gene. Exploiting this unique feature, we found that, when controlling for the number of cells in which lncRNA and protein-coding genes are expressed, lncRNAs are not less expressed, neither are more cell-type specific. Liu et al.[66] made a similar observation in brain tissue although their study was heavily constrained by the number of cells analyzed (<250 cells). This result raises the intriguing question of why lncRNAs' expression is systematically restricted to fewer cells but when transcribed they reach similar expression levels to protein-coding genes. In a recent study, Johnsson et al.[67] use allele-sensitive single-cell RNA sequencing to assess the transcriptional dynamics of lncRNAs. Their results show that lncRNAs have lowered transcriptional burst frequencies and longer duration between those bursts. Consistent with this, our previous work showed that lncRNAs harbor fewer transcription factor binding sites and higher chromatin repressive marks in their promoter regions compared to equally expressed protein-coding genes[22]. In addition, transcription factor binding sites in lncRNAs' promoters are less complex than those in protein-coding genes, suggesting that fewer transcription factors can bind to lncRNAs' promoters[23]. Overall, these results are consistent with a model in which the promoters of lncRNAs differ from those of equally expressed protein-coding genes in the probability of engaging

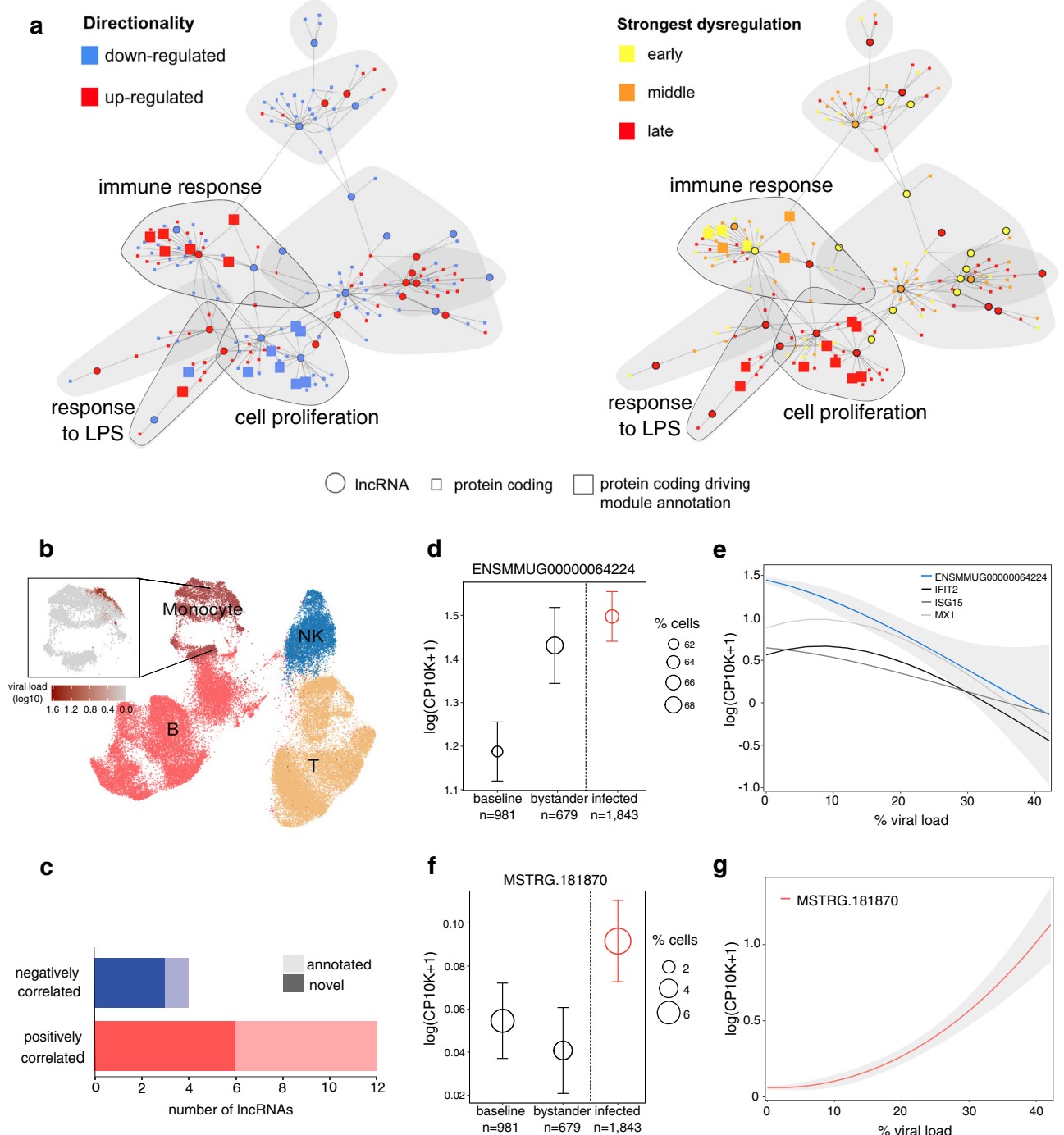

**Fig. 5 | In silico functional characterization of lncRNAs and protein-coding genes upon EBOV infection in monocytes. a** Regulatory network of lncRNAs (circles) and protein-coding (squares) DE. Vertices' colors represent up- or down-regulated genes (left) or whether a gene has the strongest fold-change compared to baseline in early (yellow), middle (orange), or late (red) stages of infection (right). Modules with significant enrichments are circled in gray and their description summarizes top enriched terms. **b** UMAP embedding of 56,317 cells from the ex vivo dataset. The magnified UMAP shows the viral load in monocytes. **c** Number of lncRNAs correlated with viral load in monocytes. **d** Expression of the lncRNA *ENSMMUG00000064224* in monocytes in baseline, bystander, and infected cells (24 h). Dots' centers represent the mean. Error bars indicate the 95% confidence interval around the mean, calculated using the standard error of the mean. Dots'-sizes represent the percentage of cells expressing the gene. *N* corresponds to the number of cells in each infection stage. **e** Expression of *ENSMMUG00000064224* and ISGs versus viral load. The shaded area around the smoothed line represents the 95% confidence interval (loess smoothing method). **f, g** Same as (**d**, **e**) for the lncRNA *MSTRG.181870*.

in active transcription rather than in the strength of the transcriptional response.

LncRNAs whose expression is condition or cell-type specific are candidate disease biomarkers and potential therapeutic targets[68]. Multiple metrics have been developed to measure tissue specificity in bulk data[41], but none of those has been specially designed to measure cell-type specificity. In this study, we introduce Upsilon, a metric that leverages the unique feature of single-cell technologies to know the number of cells expressing a gene to estimate cell-type specificity. We have identified 153 cell-type specific lncRNAs in PMBCs, including

some disease biomarkers supporting the utility of this metric to identify disease-related genes. We anticipate that, with the growing availability of single-cell transcriptomics data[69], Upsilon will be extensively used in advancing our understanding of cell-type specific processes in the context of health and disease. Furthermore, our work has consistently shown that lncRNAs have cell-type and stage-specific regulation upon EBOV infection, to a similar extent to that of protein-coding genes, including a differential response when comparing infected versus bystander monocytes. Further studies with larger sample sizes will increase our understanding of lncRNA regulation upon viral entry and immune stimulation.

Collectively, this study elucidates the roles of lncRNAs in response to EBOV infection and paves the way for future studies on how to systematically analyze lncRNAs at single-cell resolution.

## Methods

### Animal sampling

No animal handling was involved in this study. Samples from Rhesus macaques (Macaca Mulatta, 43 samples across 12 tissues) were obtained from ref. 70. Animal handling was performed in accordance with the Guide for the Care and Use of Laboratory Animals of the National Institute of Health, the Office of Animal Welfare, and the US Department of Agriculture. In addition, some other samples were obtained from commercially available samples of 2 Rhesus macaques (Macaca Mulatta, 16 samples across 10 tissues) (Zyagen, San Diego, CA, USA).

### RNA sample processing

For de novo annotation, we generated paired-end, strand-specific bulk short-read RNA-sequencing (RNA-Seq) on high-quality, commercially available rhesus macaque (*Macaca mulatta*) total RNA (Zyagen, San Diego, CA, USA; hereafter referred to as Zyagen) of non-infected samples from 10 different tissues (Supplementary Data 1). Briefly, we depleted ribosomal RNA and performed random-primed cDNA synthesis[71], followed by second strand marking and DNA ligation[72] with adapters containing unique molecular identifiers (UMIs)[73] (IDT, Coralville, IA, USA). We performed the identical bulk RNA-Seq protocol but without UMIs on rhesus macaque RNA samples from 12 different tissues from the study by Luke et al.[70]. In addition, we downloaded whole blood bulk short-read RNA-Seq data from healthy samples and samples infected with Makona Ebola Virus from the NCBI Gene Expression Omnibus (GEO; accession number GSE115785). For the single-cell RNA-Seq analysis, we downloaded the PBMCs dataset from the NCBI Gene Expression Omnibus (GEO) with accession number GSE158390.

### QC and mapping

First, we merged Ensembl Mmul_10 release 100 assembly and Ensembl release 100 gene annotation with the Ebola virus/H. sapiens-tc/COD/1995/Kikwit-9510621 (GenBank #KU182905.1; *Filoviridae: Zaire ebolavirus*) assembly and annotation, respectively, and used them throughout all downstream analyses. We used Hisat v2.1.0[74] to compute assembly indexes and known splice sites and mapped each sample's reads to the merged assembly. We ran Hisat2 with default parameters, except for RNA-strandness, which we set according to the experiments' strandness (Supplementary Data 1), previously inferred with InferExperiment.py from RSeQCc v3.0.0[75]. We sorted mapped bam files with samtools sort v1.9[76] with default parameters. We retained only paired and uniquely mapped reads using samtools view with parameters -f3 -q 60. In addition, we removed duplicates from the samples tagged with UMIs (Zyagen) (Supplementary Data 1) with umi_tools dedup v1.0.0[77]. We excluded all samples with less than 10 M sequenced reads, a mapping rate lower than 0.3, or a genic mapping rate lower than 0.7. We defined the genic mapping rate as the proportion of exonic and intronic reads, as computed by read_distribution.py from RSeQCc v3.0.0[75] (see Supplementary Data 1).

### LncRNA discovery pipeline

We ran de novo transcriptome assembly separately on each sample with Stringtie v1.3.6[78], with default parameters except for strand information that was set depending on the dataset (Supplementary Data 1). We used Stringtie to merge all the de novo assemblies using the parameter "--merge". To identify novel transcripts absent from the reference annotation, we used Gffcompare v0.10.6 and retained exclusively the transcripts with class codes "u" and "x", corresponding to intergenic and antisense transcripts. We removed mono-exonic transcripts, transcripts shorter than 200 bp, and kept only transcripts abundantly expressed (log(TPM) > 0.5) in at least three samples. To assess the coding potential of the newly assembled transcripts, we used three sequence-based lncRNAs prediction tools: Coding Potential Assessment Tool v3.0.0 (CPAT)[79], Coding Potential Calculator v2.0 (CPC2)[80], and Coding-Non-Coding Identifying Tool v2 (CNIT)[81] with default parameters. For each independent prediction tool, we removed genes with at least one isoform predicted as non-coding and one as protein-coding. We considered a gene to be a long non-coding RNA if the three tools classified it as non-coding. We then merged the obtained list of novel lncRNAs to the reference annotation and used it in downstream analyses. To benchmark our lncRNAs discovery pipeline, we predicted the biotype of annotated genes (Ensembl v100) (coding or non-coding) and compared our predictions to their annotated biotype. To compare lncRNA and protein-coding transcript length, number of exons and exon length, we considered the longest transcript per gene. To identify lncRNAs orthologs to human, we used the synteny-based lncRNAs detection tool slncky v1.0 on human hg38 assembly and gencode hg38 v23 annotation[82]. For the sake of reproducibility, the lncRNAs discovery pipeline is implemented in Nextflow[83] and combined with Singularity software containers.

### Tissue-specificity estimates

We calculated gene tissue-specificity scores using Tau[36] based on average tissue TPM gene expression values. Tau ranges from 0 to 1: genes with a score close to 1 are more specifically expressed in one tissue, while genes with a score closer to 0 are equally expressed across all tissues. We classified genes as tissue-specific (Tau > 0.7), intermediate ($0.3 \le$ Tau $\le 0.7$), or ubiquitous (Tau < 0.3). For tissue-specific genes, we determined the tissue in which they exhibited the highest average expression (log10TPM value) and considered them to be specific for that tissue. To compare the expression levels between tissue-specificity groups, we selected the expression value of the tissue with the highest average expression for each gene.

### Single-cell RNA sequencing data and processing

We used two publicly available single-cell RNA-Seq datasets of Rhesus Macaque peripheral mononuclear cells (PBMCs) infected with EBOV in vivo and ex vivo[30]. The in vivo dataset comprised samples from 21 individuals, collected before and at several days post-infection (DPI) with EBOV, and contained 38,067 cells. We performed the gene quantification using the Drop-seq analysis pipeline (https://github.com/broadinstitute/Drop-seq), with the scripts executed using Nextflow[83] and Singularity containers for better reproducibility (https://github.com/Mele-Lab/2023_SingleCellEbolaLncRNAs_NatComms). We used Scrublet v.0.2.1[84] for doublet detection and applied the IntegrateData method of Seurat v3.0[85] for fresh versus frozen batch effect correction. To select suitable filtering thresholds, we followed the best practices for single-cell analyses[86], including the selection of cells with at least 1000 and a maximum of 10,000 UMIs, at least 600 and a maximum of 2000 detected genes, and the exclusion of cells with more than 5% of

mitochondrial reads. The counts were normalized to log(CP10K + 1) after removing viral transcripts to avoid library size normalization biases.

The ex vivo dataset included PBMCs from healthy macaques, either inoculated, irradiated, or incubated with the virus, that were sequenced at 4 or 24 h post-infection (Supplementary Fig. 11) and it contained 56,317 cells. We followed the same processing steps as the in vivo dataset but increased the upper thresholds to ensure we did not exclude highly infected cells or cells with particularly increased expression of host genes, keeping those with less than 15,000 UMIs and less than 4000 detected genes per cell.

To replicate some of our observations in human data, we used available gene counts of human healthy PBMCs from 10x Genomics[40] (32,738 available cells) and human Ensembl version 100 gene annotation. We applied the same QC and filtering protocols.

## Single-cell clustering and cell type identification
To cluster cells, we used the Louvain algorithm as implemented in the Seurat package[85]. To identify cluster-specific genes, we ran a differential expression analysis between each cluster and all the remaining ones using the Seurat function FindAllMarkers. Based on the expression levels of known marker genes, we classified clusters into the four major PBMCs cell types (T cell, B cell, Monocytes, and Natural Killers) (Fig. 2a).

## LncRNA and protein-coding gene comparisons
We used Seurat's normalization values (log(CP10K + 1)) to compare expression levels between lncRNAs and protein-coding genes. We considered a gene to be expressed in a cell when its normalized expression value was larger than 1. This generated a total of 2037 lncRNA and 13,718 protein-coding genes. To compare the properties (i.e., expression or number of cells in which a gene is expressed or cell-type specificity), we only used genes expressed in more than 60 cells, leaving a total of 925 lncRNA and 11,321 protein-coding genes. Median expression values were calculated exclusively across cells in which the gene was expressed. We used the MatchIt R package v4.0.0 (https://www.rdocumentation.org/packages/MatchIt/) to obtain the pairs of lncRNA and protein-coding genes matched either by median expression or by the percentage of cells in which they were expressed.

## Cell-type specificity estimates
We considered two distinct cell-type specificity measurements. First, we leveraged Tau[41], a metric originally designed to assess tissue-specificity. Instead of calculating the mean expression per tissue for each gene, we calculated the mean expression per cell type, including zeros. Tau was calculated as follows:

$$\tau = \frac{\sum_i (1 - \hat{x}_i)}{n - 1}, i = 1, 2 \ldots n; \hat{x}_i = \frac{x_i}{\max_{i=1\ldots n}(x_i)} \tag{1}$$

where $x_i$ is the mean expression of a gene in cell type $i$ and $n$ is the total number of cell types. In addition, we designed a score (Upsilon, $v$) that relies purely on the proportion of cells in which each gene is expressed, which was calculated as follows:

$$v = \max_{j=1\ldots n} \frac{O_{i,j} - E_j}{1 - E_j} \tag{2}$$

where:

– $O_{i,j}$ is the observed proportion of cells in which gene $i$ is found expressed in cell type $j$. To calculate the proportions of cells in which each gene is expressed per cell type, we considered only the cells in

which we detected the gene as expressed, so that, per gene, the proportions assigned to the different cell types sum up to one.

– $E_{i,j}$ is the expected proportion of cells in which gene $i$ would be expressed in cell-type $j$ if it was not cell-type specific. The expected proportion of cells for cell type $j$ is equal for all the genes and corresponds to the proportion of cells of cell type $j$ in the dataset.

Then, we divided the difference between the observed and expected proportions by the maximum value this difference could reach. The maximum value is reached when the gene is expressed in all cells of one cell type, which is the difference between 1 and the expected proportion. The value, therefore, ranges from 0 to 1. We then calculated the specificity of each gene to each of the cell types and reported the maximum of these values as the gene's global specificity score.

## Cell-type specificity simulations
To explore the performance of cell-type specificity metrics, we designed different hypothetical scenarios with genes presenting three degrees of cell-type specificity (highly, intermediate, or lowly cell-type specific genes) in a cell population of three cell types. To do this, we kept a fixed expression value for expressed genes (TPM = 2) and zero for non-expressed and modified the proportion of cells of a particular cell type where the gene was expressed (50%, 30%, and 20% of the total number of cells) (Supplementary Fig. 5).

## Marker and housekeeping genes selection
We obtained the list of PBMC marker genes with the Seurat[85] function FindAllMarkers. As housekeeping genes, we selected *RRN18S, RPLPO, GAPDH, ACTB, PGK1, RPL13A, ARBP, B2M, YWHAZ, SDHA, TFRC, GUSB, HMBS, HPRT1,* and *TBP*[87–89]. We used the cell-type specificity score of the collected marker and housekeeping genes to compare the ability of Upsilon and the repurposed Tau to distinguish established cell-type specific and ubiquitous genes.

## Correlation tissue and cell-type specificity
To determine the correlation between tissue specificity and cell-type specificity, we selected genes expressed in both whole blood samples from the bulk RNA-seq dataset (average TPM > 0.1) and in the single cell in vivo PBMC dataset (log(CP10K + 1) > 1 in at least 10 cells). A total of 1532 lncRNAs and 11,501 protein-coding genes were obtained (Supplementary Fig. 6F). We then conducted a Fisher exact test to confirm that the overlap was significant. The variables tested included genes expressed in both datasets, genes expressed only in whole blood, genes expressed in PBMCs, and macaque-annotated genes not expressed. Using the resulting set of expressed genes in both datasets, we calculated the Spearman correlation coefficient separately for lncRNAs and protein-coding genes, to determine the correlation between tissue specificity Tau and cell-type specificity Upsilon.

## Differential expression analysis
We grouped samples of the in vivo dataset based on their day post-infection: baseline (0 DPI) (13 individuals), early (3 DPI) (3 individuals), middle (4–5 DPI) (4 individuals), and late stages (6–8 DPI) (8 individuals). We ran differential expression analysis using MAST v1.12.0[90] in each cell type separately. We excluded neutrophils as they were detected exclusively at later stages of infection. As input, we used the log-normalized and scaled expression counts (logCP10K + 1) from those genes expressed in at least 10% of the cells within each cell type. We performed pairwise comparisons between each stage of infection (early, middle, late) and baseline within each cell type separately. We fit a hurdle model that included as covariates the number of genes detected per cell and a binary variable corresponding to the processing of the sample, whether it was fresh or frozen. The resulting model was the following:

Expression (logCP10K + 1) ~ InfectionStage + NumDetectedGenes + SampleProcessing

Differential expression *P*-values were corrected with Benjamin and Hochberg multiple testing[91]. Genes were considered to be DE if they had a logFC > 0.1 and adjusted *P*-value < 0.05.

## Expression dynamics differences between lncRNA and protein-coding genes

We used Fisher's exact test to investigate whether lncRNAs have differential expression patterns more cell-type-specific or stage-specific than protein-coding genes. The two tested variables are gene biotype and whether the gene is DE in one or more cell types or the stage.

## Gene colocation analysis

We used the GenomicRanges package v1.38.0 (https://bioconductor.org/packages/release/bioc/html/GenomicRanges.html) to calculate the genomic distance between genes in the macaque Ensembl v100 annotation. We considered a pair to be co-located if they are less than 1 Mbp. To test whether DE lncRNAs were closer to DE protein-coding genes more often than not DE lncRNAs, we set up Fishers' exact test. The two tested variables were whether the lncRNA is DE and whether it is in cis to a DE protein-coding gene.

## Co-expression network

We built a co-expression network using all differentially expressed genes in monocytes with GrnBoost2[92]. To focus on the co-regulatory network involving lncRNAs, we only retained edges connected to at least one lncRNA. Also, we retained the top 0.5% edges when sorted by weight. We identified communities with the Louvain algorithm[93] and reported those with at least 7 edges. For the functional enrichment of the modules, we used the R package clusterProfiler v4.2.0[94].

## Correlation with viral load

To determine the correlation between viral transcript changes and gene expression in infected cells, we focused solely on monocytes at a late stage of infection (24 h post-infection ex vivo and 6–8 days post-infection in vivo). We obtained the viral load by dividing the number of viral counts by the total number of counts and then computed the Spearman correlation coefficient between the viral load (log10) and the normalized expression of each gene (log(CP10K + 1)). The resulting *P*-values were corrected for multiple testing using the Benjamin and Hochberg method[91].

## Ethics

The study was performed in accordance with the Guide for the Care and Use of Laboratory Animals of the National Institute of Health, the Office of Animal Welfare, and the US Department of Agriculture[38].

## Reporting summary

Further information on research design is available in the Nature Portfolio Reporting Summary linked to this article.

## Data availability

The sequencing data generated in this study have been deposited in the NCBI Gene Expression Omnibus (GEO) database under accession code GSE192447. The publicly available whole blood bulk short-read RNA-Seq data from healthy samples and samples infected with Makona Ebola Virus data used in this study are available in the NCBI Gene Expression Omnibus (GEO) database under accession code GSE115785. The single-cell RNA-Seq data used in this study are available in the NCBI Gene Expression Omnibus (GEO) database under accession code GSE158390. Raw Seurat Objects for both single-cell datasets used in this study are available at Zenodo. The full co-expression network file is also provided (https://doi.org/10.5281/zenodo.7997135). The reference genome of EBOV used in this study is available in the GenBank

database under accession code KU182905.1. The assembly and reference genome of Macaca Mulatta used in this study are available in the Ensembl database (Mmul_10) (https://ftp.ensembl.org/pub/release-100/fasta/macaca_mulatta/dna/Macaca_mulatta.Mmul_10.dna.toplevel.fa.gz, https://ftp.ensembl.org/pub/release-100/gtf/macaca_mulatta/Macaca_mulatta.Mmul_10.100.gtf.gz). The assembly and reference genome of human used in this study are available in the Gencode database (release_23) (https://ftp.ebi.ac.uk/pub/databases/gencode/Gencode_human/release_23/gencode.v23.annotation.gtf.gz, https://ftp.ebi.ac.uk/pub/databases/gencode/Gencode_human/release_23/GRCh38.primary_assembly.genome.fa.gz). Source data are provided with this paper.

## Code availability

The code used for this study is available at: https://github.com/Mele-Lab/2023_SingleCellEbolaLncRNAs_NatComms.

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

## Acknowledgements

We thank Kaia Mattioli for her thoughtful comments on the manuscript and Aida Ripoll-Cladellas for feedback and helpful discussions. This material was based upon work supported by Grant RYC-2017-22249 funded by MCIN/AEI/10.13039/501100011033 and Grant PID2019-107937GA-I00 funded by MCIN/AEI/10.13039/501100011033 (M.M.), the Howard Hughes Medical Institute Investigator Award (P.C.S.), the National Institute of Allergy and Infectious Diseases (NIAID) U19AI110818, the US Food and Drug Administration (FDA) contract HHSF223201810172C. We acknowledge SAB Biotherapeutics as partners for providing study materials from the study by Luke et al.[70] and for their collaborative support that allowed the study's success. Figures 1a, 1c, 4a and Supplementary Fig. 11A were created with BioRender.com.

## Author contributions

L.S. and M.S.R. performed the computational analysis. M.M. designed the project. J.L.R. contributed to the study design. M.M. and R.G.P. supervised the analysis. L.S., M.S.R., M.M., and R.G.P. wrote the manuscript. A.E.L., G.C.A., K.G.B., K.J.S., and S.W. did all the experimental work. F.R. contributed to the design of the cell-type specificity score. L.E.H., R.S.B., and P.C.S. designed and led all experimental work. All authors have read and approved the manuscript for publication.

## Competing interests

SAB Biotherapeutics, Inc. provided the study materials from the study by Luke et al. None of the authors of this study has financial interest in SAB Biotherapeutics, Inc. company. P.C.S. is a co-founder of, shareholder in, and advisor to Sherlock Biosciences, Inc.; a board member of and shareholder in the Danaher Corporation; and a co-founder of and shareholder in Delve Bio. The other authors declare no competing interests.
