## [Peer Review File · Nature Communications]

Single-cell profiling of lncRNA expression during Ebola virus infection in rhesus macaquesREVIEWER COMMENTS

Reviewer #1 (Remarks to the Author):

Long noncoding RNA genes are the largest, yet the most enigmatic component of our genome. Although their growing number is linked with fundamental biological and various pathophysiological processes, the vast majority (>97%) of human lncRNAs remain to be functionally characterised. Despite the advancements, our understanding of the lncRNA sequence-function relationship is poor. Also the knowledge about the sequence similarity of lncRNAs across species is very limited, which largely hampers their identification and functional characterisation. Therefore, reliable assignment of biological roles requires extensive studies of lncRNAs in other mammalian and vertebrate species, particularly in specific biological and medical contexts.

The authors investigate lncRNA expression at single-cell resolution during Ebola virus (EBOV) infection using rhesus macaque. First, they expand lncRNA annotation by assembling nearly 4,000 novel lncRNAs from bulk RNA sequencing data derived from both infected and not infected samples. Next, they performed through analysis of lncRNAs using single-cell expression data from macaque's peripheral blood mononuclear cells. Finally, the authors introduce – Upsilon – a new metric to estimate the cell-type specificity at single-cell resolution.

This is a very interesting study, which shows that lncRNAs are on average expressed in fewer cells than protein coding genes, which can explain their overall expression levels. On the other hand, lncRNAs and protein coding expressed in a similar number of cells, display similar expression levels. The data are compelling, whereas the paper itself can be of interest for broader scientific communities. However, the authors might address a couple of points to clarify some aspects of their work:

1. The description of the novel lncRNA models could be improved. It should be clearly stated that these models were assembled from short-read RNA-seq data (please add this information to the Materials and Methods section and highlight it also in the main text). Application of short-read RNA-seq data in the context of gene annotation is still not trivial, as the necessary compromise between read length and sequencing depth severely hampers accurate full-length isoform discovery. Moreover, human lncRNA annotations also remain largely incomplete (e.g. study cited as ref. 36). Thus, comparing novel, rather fragmentary lncRNA models to the incompletely annotated ones allows only to conclude that the novel are no worse than the existing ones, but it does not mean they are full-length. Actually, the two-exon gene bias reported by Derrien et al. (cited as ref. 16) turned out to be an artefact arising from annotation incompleteness. This was clarified by another study from this group (Lagarde et al. 2017), which also showed that spliced length of full-length lncRNAs and protein coding genes is comparable, as lncRNAs have fewer, but on average longer exons. Moreover, Derrien et al. reported no major difference in the length of introns for lncRNAs and protein coding genes. This could be another way of comparing the assembled, novel lncRNAs to the existing ones. Finally, it would be interesting to know what contributes to the difference in the exons lengths for macaque transcripts (lncRNA vs. protein coding and macaque vs. human lncRNAs).

2. Analysis of syntenic lncRNA orthologues between human and macaque is very interesting. However, it requires additional explanation on how the orthologue length comparison can confirm that the pipeline detects full-length transcripts. Syntenic lncRNA orthologues have highly variable transcript structures across species. Moreover, it seems that they undergo distinct processing that affects their functional potential. As shown by Guo et al. (Cell, 2020) this effect is rather species not cell type specific.

3. The conclusion that novel lncRNAs are more tissue specific is interesting. It would be good to know what is the average expression of tissue-specific lncRNAs in comparison to

group-enriched and ubiquitously expressed ones. Also a plot showing (in absolute numbers) distribution of tissue-specific lncRNAs across tissues would nicely complement the existing analysis.

4. The observation that lncRNAs are consistently expressed in fewer cells sounds very intriguing. I was wondering if this analysis could be extended by distinguishing the tissue-specific and group-enriched lncRNAs. Such analysis could reveal whether the tissue-specific lncRNAs have more reliable, but restricted expression patterns (are consistently expressed in cells of specific types) or are rather more likely to be the non-functional transcriptional noise.

5. Upsilon seems to nicely differentiate the cell-type specific and ubiquitous genes. It is worth checking if the genes (lncRNA and pc) with high and low Upsilon values have any specific genomic features that could additionally distinguish these classes.

Minor comments:

- Figure 1E does not seem to be referenced in the main text.
- Figure 4C – please consider putting the number of DE under each heat map. Please also explain in the legend, why some records have their gene names displayed in the figure.
- Please describe the link between figure 4D and S7A.
- Figure S3A – the low contrast between colours for novel and annotated lncRNA makes the figure difficult to interpret.
- Figure S7A, please consider using transparency to show the number of novel and annotated genes.
- Figure S7C and D – blue and read colours were used to highlight the protein coding and lncRNAs, respectively. I suggest sticking with the fixed colour-code. For the same reason I suggest changing the colours for infected/non-infected samples.
- The distribution of the figures is forcing the reader to constantly switch between main and supplementary figures. This could be easily optimised. For example, Figure 4E and S7E could be combined into a single plot/panel.

Reviewer #2 (Remarks to the Author):

In the submitted manuscript, Santus et al have provided a comprehensive resource aiming to catalog long non-coding RNA (lncRNA) molecules from healthy and Ebola-infected rhesus macaques. The authors started by gathering all existing RNA-seq data (ref #33) to uncover novel lncRNA revealing close to 4000 new lncRNA (Fig.1). Then they leveraged single-cell RNA-seq data from healthy macaque (ref #30) to uncover the cell-type specificity of the lncRNAs (Fig.2&3). Finally, they analyzed how lncRNAs are regulated under Ebola infection (Figure 4&5).

lncRNAs play a pivotal role in the regulation of biological processes. Overall, this publication is leveraging nicely published datasets to discover novel lncRNAs. The number of newly discovered new lncRNAs is impressive and this study will become a major resource for scientists studying rhesus macaque and Ebola infection. From an infection point-of-view, the resource can seed many novel discoveries.

However, I have some major comments :

- When looking at the blood single-cell RNA-seq, why only 925 lncRNA could be detected (Fig 2A)? How many lncRNA could be uncovered using blood bulk RNA-seq? How do both data match?

- Since lncRNAs have a specific expression pattern among blood cells, is it possible that the authors make a UMAP of blood cells using lncRNA only in Figure 3?

- The authors discuss extensively the expression features of lncRNA. They claim that lncRNA are expressed at similar levels to protein-coding genes (Fig. 2A). This is possible but the authors should explain why this result is in contradiction with the report from Sandberg and co (PMID: 35241826) that shows a massive discrepancy in expression level between lncRNA and protein-coding genes. I would advise validating the results using smFISH experiments.

- It is tempting to ask for more experimental work on the identified lncRNA specific for infected cells but working with Ebola virus requires a huge effort in the BSL4 lab. This certainly goes beyond the scope of this publication.

Minor comments:

-Suppl. Tables should have been provided in an Excel table.

-In Table S1: add the reference of the publications with a PMID

-Please add the Figure number on top of each figure

-Suppl Figure 11 is not quoted – maybe line 306 – there is a typo.

We would like to thank the reviewers for their valuable feedback. We are extremely excited about the new analyses and results we have generated. We have done many novel analyses that have expanded our previous findings and, in some cases, the received comments have allowed us to approach our initial questions from another perspective. Overall, this has significantly improved our work and has generated a more complete manuscript. Replies to the reviewer's comments are outlined below (in blue) and all changes to the Manuscript are highlighted (in blue).

Reviewer 1:

Long noncoding RNA genes are the largest, yet the most enigmatic component of our genome. Although their growing number is linked with fundamental biological and various pathophysiological processes, the vast majority (>97%) of human lncRNAs remain to be functionally characterised. Despite the advancements, our understanding of the lncRNA sequence-function relationship is poor. Also the knowledge about the sequence similarity of lncRNAs across species is very limited, which largely hampers their identification and functional characterisation. Therefore, reliable assignment of biological roles requires extensive studies of lncRNAs in other mammalian and vertebrate species, particularly in specific biological and medical contexts.

The authors investigate lncRNA expression at single-cell resolution during Ebola virus (EBOV) infection using rhesus macaque. First, they expand lncRNA annotation by assembling nearly 4,000 novel lncRNAs from bulk RNA sequencing data derived from both infected and not infected samples. Next, they performed through analysis of lncRNAs using single-cell expression data from macaque's peripheral blood mononuclear cells. Finally, the authors introduce – Upsilon – a new metric to estimate the cell-type specificity at single-cell resolution.

This is a very interesting study, which shows that lncRNAs are on average expressed in fewer cells than protein coding genes, which can explain their overall expression levels. On the other hand, lncRNAs and protein coding expressed in a similar number of cells, display similar expression levels. The data are compelling, whereas the paper itself can be of interest for broader scientific communities.

We appreciate the reviewer's encouraging words about our work.

However, the authors might address a couple of points to clarify some aspects of their work:

1. The description of the novel lncRNA models could be improved. It should be clearly stated that these models were assembled from short-read RNA-seq data (please add this information to the Materials and Methods section and highlight it also in the main text). Application of short-read RNA-seq data in the context of gene annotation is still not trivial, as the necessary compromise between read length and sequencing depth severely hampers accurate full-length isoform discovery.

Moreover, human lncRNA annotations also remain largely incomplete (e.g. study cited as ref. 36). Thus, comparing novel, rather fragmentary lncRNA models to the incompletely annotated ones allows only to conclude that the novel are no worse than the existing ones, but it does not mean they are full-length. Actually, the two-exon gene bias reported by Derrien et al. (cited as ref. 16) turned out to be an artefact arising from annotation incompleteness. This was clarified by another study from this group (Lagarde et al. 2017), which also showed that spliced length of full-length lncRNAs and protein coding genes is comparable, as lncRNAs have fewer, but on average longer exons. Moreover, Derrien et al. reported no major difference in the length of introns for lncRNAs and protein coding genes. This could be another way of comparing the assembled, novel lncRNAs to the existing ones. Finally, it would be interesting to know what contributes to the difference in the exons lengths for macaque transcripts (lncRNA vs. protein coding and macaque vs. human lncRNAs).

We absolutely agree with the reviewer's comment regarding the limitations of using short-read data to perform *de novo* annotation. We have rewritten the corresponding methods, results, and discussion sections to emphasize these limitations.

Specifically, we have made the following changes:

1. We have explicitly mentioned the use of short-read data:

- 1.1. In the **Main** text:

“To improve the current lncRNA annotation, we generated short-read RNA-sequencing data from 13 tissues”

- 1.2. In the **Methods** section:

“For de novo annotation, we generated paired-end, strand-specific bulk short-read RNA-sequencing (RNA-Seq).”

2. We have removed the statement about capturing full-length reads:

“Considering the difficulties of reconstructing transcript structures from short-read data, we leveraged the human annotation to assess the quality of our de novo annotated transcripts.”

Also, we agree with the reviewer that the generation of long-read datasets may help refine lncRNA annotations even further. Future work using long-read sequencing on non-model organisms such as macaque in the context of infection and immune activation will help bridge this gap. We have mentioned it in the **Discussion**:

“Future work using emerging long-read sequencing technologies⁶⁵ will further improve the discovery and annotation of lncRNAs in model species in the context of infection.”

In addition, we observe that novel lncRNAs are significantly shorter than mRNAs but similar in length to annotated human lncRNAs. Thus our newly developed pipeline identifies novel lncRNAs with similar properties to current approaches used by the Gencode team¹ annotating lncRNAs in humans.

Also, as suggested by the reviewer we have computed intron length and added the plot as **Supplemental Figure S2A**:

Supplementary Fig. 2A. Distribution of intron length in novel and annotated macaque and human intergenic and antisense lncRNAs and protein-coding genes.

2. Analysis of syntenic lncRNA orthologues between human and macaque is very interesting. However, it requires additional explanation on how the orthologue length comparison can confirm that the pipeline detects full-length transcripts. Syntenic lncRNA orthologues have highly variable transcript structures across species. Moreover, it seems that they undergo distinct processing that affects their functional potential. As shown by Guo et al. (Cell, 2020) this effect is rather species not cell type specific.

We apologize for not being more precise in this statement. Our intention was to show that our novel annotated transcripts were as good quality as those annotated by the Gencode team¹ in humans by comparing macaque and human ortholog lengths. Transcript length will of course vary between human and macaque orthologs but if the annotations are of equal quality, then macaque transcripts should not be systematically shorter. That is what we tested. We observe random variation in length between orthologs, as expected, but we do not see a bias. Thus, the novel lncRNAs are not systematically shorter than their human orthologs suggesting that the quality of our novel lncRNA is as good as the current human lncRNA annotation.

We have clarified this in the main text and now it reads as follows:

“Considering the difficulties of reconstructing transcript structures from short-read data, we leveraged the human annotation to assess the quality of our de novo annotated transcripts. We reasoned that if our approach recovered high-quality transcripts, we should not observe any length bias towards shorter transcripts when comparing our set of lncRNAs to their human orthologs. Thus, we selected all novel lncRNAs that had a human ortholog (see Methods) and tested if the transcript length of macaque novel lncRNAs were shorter than the human orthologs. We observed that macaque lncRNAs were not systematically shorter than their human counterparts. (One-tailed paired Wilcoxon signed-rank test, P -value > 0.05) (Supplemental Fig. 2E).”

3. The conclusion that novel lncRNAs are more tissue specific is interesting. It would be good to know what is the average expression of tissue-specific lncRNAs in comparison to group-enriched and ubiquitously expressed ones. Also a plot showing (in absolute numbers) the distribution of tissue-specific lncRNAs across tissues would nicely complement the existing analysis.

We thank the reviewer for this insightful comment. For this supplementary analysis, we used the same classification as presented in the original manuscript which leverages the tissue-specificity metric Tau to label genes as tissue-specific ($\text{Tau} > 0.7$), intermediate ($0.3 \leq \text{Tau} \leq 0.7$), and ubiquitous ($\text{Tau} < 0.3$). The results are depicted in the plot below.

Fig 1G. Bar plot showing the percentage of ubiquitous ($\text{Tau} < 0.3$), intermediate ($0.3 \leq \text{Tau} \leq 0.7$), and tissue-specific ($\text{Tau} > 0.7$) lncRNAs and protein-coding genes. Labels indicate the number of genes within each category

Additionally, as suggested, we added a plot of the distribution across tissues of novel and annotated tissue-specific lncRNAs. To do so, we determined the tissue in which they exhibited the highest average expression ($\log_{10}\text{TPM}$ value) and considered them to be specific for that tissue. We detect a high number of tissue-specific lncRNAs in whole blood and ovary likely due to the higher number of samples in our dataset.

Supplementary Fig 3C. Bar plot showing the distribution of novel (top) and annotated (bottom) tissue-specific lncRNAs ($\text{Tau} > 0.7$) across tissues.

Then, to compare the expression levels between the three tissue-specificity groups, we selected for each gene, the average expression of the tissue where the value was the highest. While ubiquitous novel and annotated lncRNAs had similar expression levels, novel tissue-specific and intermediate lncRNAs were more expressed than annotated lncRNAs. See the plot below.

Fig. 1F. Distribution of average expression ($\log_{10}\text{TPM}$) in the tissue with the highest expression of tissue-specific ($\text{Tau} > 0.7$), intermediate ($0.3 \leq \text{Tau} \leq 0.7$), and ubiquitous ($\text{Tau} < 0.3$) lncRNAs and protein-coding genes

We think both analyses significantly add to the manuscript and we have included them in the new version.

4. The observation that lncRNAs are consistently expressed in fewer cells sounds very intriguing. I was wondering if this analysis could be extended by distinguishing the tissue-specific and group-enriched lncRNAs. Such analysis could reveal whether the tissue-specific lncRNAs have more

reliable, but restricted expression patterns (are consistently expressed in cells of specific types) or are rather more likely to be the non-functional transcriptional noise.

The reviewer raises a very interesting point. To address this, we performed a correlation analysis between tissue-specificity and cell-type specificity comparing Tau and Upsilon values respectively. We find a significant correlation between tissue and cell-type specificity (Spearman $\rho=0.31$, p-value $< 2.2 \times 10^{-16}$), implying that tissue-specific lncRNAs exhibit more cell-type restricted expression patterns. Consistently the same pattern is observed for protein-coding genes² (Spearman $\rho=0.46$, p-value $< 2.2 \times 10^{-16}$). Results are shown in the plots below, which have been added as **Figure 3C** and **Supplementary Figure 6G**.

Fig. 3C and Suppl. 6G. Distribution of tissue-specificity metrics in ubiquitous ($Upsilon < 0.3$), intermediate ($0.3 \leq Upsilon \leq 0.7$) and cell-type specific ($Upsilon > 0.7$) lncRNAs (left) and protein-coding genes (right).

5. Upsilon seems to nicely differentiate the cell-type specific and ubiquitous genes. It is worth checking if the genes (lncRNA and pc) with high and low Upsilon values have any specific genomic features that could additionally distinguish these classes.

Following the reviewer's suggestions, we have conducted an evaluation of the genomic features of the three classes identified by Upsilon (ubiquitous, intermediate and cell-type specific). The results are presented in the plots below. We found that cell-type-specific lncRNAs have slightly shorter transcript lengths and slightly fewer and shorter exons as compared to ubiquitous genes (Mann-Whitney U test, all P-values $< 3.5 \times 10^{-3}$). The plots are in **Supplemental Figure S6C-E**.

Suppl. 6C-E. Distribution of transcript length, exon length, and number of exons per transcript in ubiquitous, intermediate, and specific lncRNA and protein-coding genes

Minor comments:

- Figure 1E does not seem to be referenced in the main text.

It was referenced in lines 109-110 (Figure 1D-E) in the original manuscript after the following statement:

“As expected, novel and annotated lncRNA transcripts were shorter and had longer and fewer exons compared to protein-coding genes (Mann-Whitney U test, all P-values < 2.2 x10⁻¹⁶) (Figure 1D-E).”

- Figure 4C – please consider putting the number of DE under each heat map. Please also explain in the legend, why some records have their gene names displayed in the figure.

Thank you for pointing this out. We have added the number of DE under each heatmap and clarified it in the figure capture.

- Please describe the link between figure 4D and S7A.

In Figure 4D, now Figure 4F, we are showing the overlap of lncRNA DE between the different cell types in the main plot and the number of total DE lncRNA per cell type in the side barplot. In Figure S7A, now Figure S8A, we want to illustrate whether the lncRNA differentially expressed in each cell type corresponds to a novel or annotated gene.

- Figure S3A – the low contrast between colours for novel and annotated lncRNA makes the figure difficult to interpret.

The reviewer is right. To avoid any confusion, we increased the color intensity for novel lncRNA and filled the boxplots.

- Figure S7A, please consider using transparency to show the number of novel and annotated genes.

As suggested by the reviewer, we have modified figure S7A to highlight the numbers of novel and annotated lncRNA DE using transparency. See the new figure Below:

- Figure S7C and D – blue and red colours were used to highlight the protein coding and lncRNAs, respectively. I suggest sticking with the fixed colour-code. For the same reason I suggest changing the colours for infected/non-infected samples.

We agree with the reviewer and we have modified the color code for up-/downregulated genes in **Supplemental Figure S8C-D**:

- The distribution of the figures is forcing the reader to constantly switch between main and supplementary figures. This could be easily optimised. For example, Figure 4E and S7E could be combined into a single plot/panel.

Following the reviewer’s suggestion, we have redistributed some figures while trying to keep a balance as our main panels had already several plots. We have made the following changes:

- In the section “*De novo annotation largely expands the rhesus macaque non-coding transcriptome*”, we have redistributed figure citations so supplemental figures are cited together.
- In the section “*LncRNAs are systematically expressed in fewer cells compared to protein-coding genes*” now entitled, we have reduced the number of figures in the Supplemental Figure while still conveying the message.
- In the section “*Upsilon, a novel metric to measure cell-type specificity in single-cell expression data*” and the new section “*Higher specificity of lncRNAs can be attributed to their expression in fewer cells*” we have extensively modified Figure 3 and Supplemental Figures S6 and S7 due to the additional analyses. For that, we have taken into consideration the reviewer’s suggestion.
- In the section “*LncRNAs are dynamically regulated upon EBOV infection*” we have merged Figure 4E and S7E in Figure 4E as suggested by the reviewer.
- Finally, in the section “*Functional characterization of lncRNAs differentially expressed upon EBOV infection*” we have integrated Figure 5A and Supplemental Figure S9A.

Reviewer 2

In the submitted manuscript, Santus et al have provided a comprehensive resource aiming to catalog long non-coding RNA (lncRNA) molecules from healthy and Ebola-infected rhesus macaques. The authors started by gathering all existing RNA-seq data (ref #33) to uncover novel lncRNA revealing close to 4000 new lncRNA (Fig.1). Then they leveraged single-cell RNA-seq data from healthy macaque (ref #30) to uncover the cell-type specificity of the lncRNAs (Fig.2&3). Finally, they analyzed how lncRNAs are regulated under Ebola infection (Figure 4&5).

lncRNAs play a pivotal role in the regulation of biological processes. Overall, this publication is leveraging nicely published datasets to discover novel lncRNAs. The number of newly discovered new lncRNAs is impressive and this study will become a major resource for scientists studying rhesus macaque and Ebola infection. From an infection point-of-view, the resource can seed many novel discoveries.

We thank the reviewer very much for these kind words.

However, I have some major comments:

1. When looking at the blood single-cell RNA-seq, why only 925 lncRNA could be detected (Fig 2A)?

We apologize to the reviewer because this was not described clearly enough in the Methods section. We used 925 lncRNAs in Figure 2A-G and Figure 3D-F because we are very stringent in our set of lncRNAs used for downstream analysis. However, we did identify more lncRNAs expressed.

Specifically, we followed the state-of-the-art QC filtering described in ³ and detected 2,037 lncRNAs expressed ($\log_{CP10K+1} > 1$) in at least ten cells. However, we implemented additional filtering to only consider lncRNAs expressed in at least 60 cells to ensure our results were not an artifact of transcriptional noise. This filter was used in the sections of the manuscript where we analyzed all cells together. Specifically, the sections “*LncRNAs are systematically expressed in fewer cells compared to protein-coding genes*” and “*Higher specificity of lncRNAs can be attributed to their expression in fewer cells*”. All this now is clearly specified in the Methods section. For consistency, in the remaining sections where the analyses were performed per cell type, we decided to use similar filtering criteria but at the cell-type level. Specifically, we only analyze genes expressed in at least 10% of the cells within each cell type (sections “*LncRNAs are dynamically regulated upon EBOV infection*” and “*Functional characterization of lncRNAs differentially expressed upon EBOV infection*”). This additional filter does not change our previous findings but Figures 4 and 5 have undergone some slight modifications. All this now is clearly specified in the Methods section.

2. How many lncRNA could be uncovered using blood bulk RNA-seq? How do both data match?

That is a very interesting point. Following the reviewer's suggestion, we compared the number of lncRNAs expressed in bulk whole-blood RNA-seq data (average TPM > 0.1) and single-cell RNA-seq PBMC data (logCP10K+1 > 1 in at least 10 cells). The resulting numbers are 3,898 lncRNA and 16,447 protein-coding genes in whole blood and 2,037 lncRNA and 13,718 protein-coding genes in PBMCs. Of those, 1,532 lncRNAs (35%) and 11,501 protein-coding genes (70%) are expressed in both datasets, as shown in the figure below. Both overlaps are statistically significant (Fisher exact test, p-value < 2.2e-16). The Venn Diagram below has been added as **Supplemental Figure S6B**.

Suppl. 6F. Venn Diagram of the overlap between expressed lncRNA (red) and protein-coding (blue) genes in single-cell PBMC and whole-blood bulk RNA-seq datasets

As expected we find more genes expressed using whole-blood RNA-seq data compared to single-cell given the higher sequencing depth for bulk RNA-sequencing⁴ and a larger number of cell types present in whole blood compared to PBMCs⁵. Furthermore, higher overlap in protein-coding genes is expected due to their lower specificity and higher overall expression⁶.

To further explore the relationship between the observations made in bulk and single-cell RNAseq, we decided to test the relationship between tissue-specific measurements (defined in bulk) and cell-type-specific measurements (defined in single-cell). We selected the genes expressed in both datasets and compared their tissue-specificity (measured by Tau) and cell-type specificity (measured by Upsilon). Tissue and cell-type specificity were significantly correlated both for lncRNAs (Spearman $\rho = 0.31$, p-value < 2.2×10^{-16}) and for protein-coding genes (Spearman $\rho = 0.46$, p-value < 2.2×10^{-16}). These results suggest that tissue-specific lncRNAs detected in bulk will likely exhibit more cell-type restricted expression patterns.

3. Since lncRNAs have a specific expression pattern among blood cells, is it possible that the authors make a UMAP of blood cells using lncRNA only in Figure 3?

We thank the reviewer for this suggestion, it is indeed interesting. The suggested UMAP is depicted below and shows that lncRNAs are effectively separating different cell types. The plot has been added as **Supplementary Figure 7A**.

Suppl. 7A. UMAP embedding of 38,067 cells. Cell types are indicated by the different colors.

4. The authors discuss extensively the expression features of lncRNA. They claim that lncRNA are expressed at similar levels to protein-coding genes (Fig. 2A). This is possible but the authors should explain why this result is in contradiction with the report from Sandberg and co (PMID: 35241826) that shows a massive discrepancy in expression level between lncRNA and protein-coding genes. I would advise validating the results using smFISH experiments.

We thank the reviewer for bringing this to our attention. We agree with the reviewer that when we compare general expression differences between lncRNAs and mRNAs (without any matching), lncRNAs have significantly lower expression levels but differences are modest as opposed to the large differences reported by Sandberg and colleagues. We think this discrepancy arises from how the median lncRNA and protein-coding gene expression is computed. In our analysis, as explained in the Methods section “LncRNA and protein-coding gene comparisons”, we only calculate mean/median expression level considering cells in which the gene is expressed ($\log_{CPK10} > 1$). Thus, we exclude cells with zero expression values. We did it like this because we precisely wanted to address whether lncRNAs were highly expressed in a low number of cells or whether they were lowly expressed in many cells. In the Johnsson et al. study, although it is not explicitly stated how the mean expression calculation is performed, we checked their code and we observed that cells with zero expression values are included to calculate mean expression. In the plot below we show differences in mean gene expression between lncRNA and protein-coding genes with our data when cells with zero expression values are included. When calculating mean expression levels like this, we are able to reproduce the large differences observed by Sanberg and colleagues:

Extended Fig. 1. Distribution of mean gene expression levels (\log_{CP10K}) of lncRNA (red) and protein-coding (blue) genes when cells with zero expression values are included

In addition, in the **Discussion** section, we discuss that results from Johnsson et al.⁷ reporting lowered transcriptional burst frequencies and longer duration between those bursts are consistent with our observations.

5. It is tempting to ask for more experimental work on the identified lncRNA specific for infected cells but working with Ebola virus requires a huge effort in the BSL4 lab. This certainly goes beyond the scope of this publication.

We absolutely agree with the reviewer.

Minor comments:

- Suppl. Tables should have been provided in an Excel table.

The original tables were in Excel format when the manuscript was submitted. However, they have probably been modified in the process. We have created the following link to access the tables in Excel format:

<https://drive.google.com/drive/folders/1ffzKvRrUyBwUAcbpBkwqAK2LKcUzY1SW?usp=sharing>

-In Table S1: add the reference of the publications with a PMID

We have added the PMID to the tables

-Please add the Figure number on top of each figure.

We have added Figure numbers on top of each figure.

-Suppl Figure 11 is not quoted – maybe line 306 – there is a typo.

It was indeed a typo, we have fixed it.

References

1. Chen, J. *et al.* Evolutionary analysis across mammals reveals distinct classes of long non-coding RNAs. *Genome Biol.* **17**, 1–17 (2016).
2. Karlsson, M. *et al.* A single-cell type transcriptomics map of human tissues. *Sci Adv* **7**, (2021).
3. Luecken, M. D. & Theis, F. J. Current best practices in single-cell RNA-seq analysis: a tutorial. *Mol. Syst. Biol.* **15**, e8746 (2019).
4. Wu, A. R. *et al.* Quantitative assessment of single-cell RNA-sequencing methods. *Nat. Methods* **11**, 41–46 (2013).
5. He, D. *et al.* Whole blood vs PBMC: compartmental differences in gene expression profiling exemplified in asthma. *Allergy Asthma Clin. Immunol.* **15**, 67 (2019).
6. Cabili, M. N. *et al.* Integrative annotation of human large intergenic noncoding RNAs reveals global properties and specific subclasses. *Genes Dev.* **25**, 1915–1927 (2011).
7. Johnsson, P. *et al.* Transcriptional kinetics and molecular functions of long noncoding RNAs. *Nat. Genet.* **54**, 306–317 (2022).

REVIEWER COMMENTS

Reviewer #1: please see attachment

Reviewer #2 (Remarks to the Author):

Santus et al have charted extensively lncRNAs in macaques. The publication will be a major resource for the community in the coming years. In the revised version, Santus et al have very nicely elevated my concerns regarding the discrepancy between their analysis and the publication from Sandberg and co (PMID: 35241826).

Minor comments:

- I would remove claims for novelty “upsilon, a novel metric”.
- Figure 4C: does it worth to add the baseline expression too in the heatmap?

I would like to thank the authors for their hard work in implementing reviewers comments and suggestions. This common effort has nicely complemented and elevated the initial analysis. The proposed paper has a potential to become a useful resource for lncRNA studies at single-cell resolution in various biological contexts.

There are still a few aspects that authors might want to improve prior to the publication.

1. Although I very much appreciate the explanation about the application of syntenic lncRNAs in the length-based quality assessment of newly annotated lncRNAs, I do not think it is the best approach. As mentioned before, syntenic lncRNAs represent lncRNAs with preserved genomic positions that do not necessarily show conservation at the primary sequence level. They are actually characterised by large variations at the sequence level, including overall transcript length and the number of exons. For example, positionally conserved *CASC15* lncRNA that is located in the proximity of the *SOX4* gene in the human, mouse and zebrafish genomes has highly variable transcript structure in each of these species, which largely affects its length. The genomic length of *CASC15* lncRNA in human, mouse and zebrafish is ~531, ~24 and ~21kb, respectively. Thus, the quality annotation assessment using syntenic lncRNAs across different species, even closely related ones can be misleading. Positional conservation also covers *cis*-acting lncRNAs, which can regulate neighbouring in a sequence independent way, through the act of transcription itself. Instead, I propose to simply compare the novel lncRNA genes/transcripts to the annotated ones in the macaque genome (Figure 1 and S2).
2. I am very grateful for including the analysis of intron length (Figure S2A). However, the comparison would be more meaningful and straightforward, if the lncRNA classes match the ones presented in the Figure 1D-E. In this way it is hard to directly compare newly annotated transcripts, their exons and introns to the annotated ones. Alternatively, distinguishing intergenic and antisense lncRNA classes among the annotated ones in Figure S2 would also help. Nevertheless, it looks like on average the novel transcripts are slightly shorter than the annotated ones, which I believe is expected and ok. Finally, if for some reason, the authors decide to keep the transcript length comparison using syntenic lncRNAs, the Figure S2E should also include annotated macaque lncRNA orthologues.
3. Please add the absolute numbers to Figure S1C and D.
4. Please check the citation of Figure S6E, as presented numbers do not match the ones on the Figure. I think they actually refer to Figure 3B. Again, having the numbers displayed on the plot would help a lot.
5. For clarity reasons, can you please label the panels in Figure S7 to show which or them refer to human and macaque?

We would like to express our gratitude to the reviewers for recognizing the effort we put into enhancing the manuscript and into addressing their comments and suggestions. Replies to the reviewer's comments are outlined below (in blue) and all changes to the Manuscript are highlighted (in blue).

Reviewer 1:

I would like to thank the authors for their hard work in implementing reviewers comments and suggestions. This common effort has nicely complemented and elevated the initial analysis. The proposed paper has a potential to become a useful resource for lncRNA studies at single-cell resolution in various biological contexts.

We appreciate the positive comments of the reviewer.

There are still a few aspects that authors might want to improve prior to the publication.

1. Although I very much appreciate the explanation about the application of syntenic lncRNAs in the length-based quality assessment of newly annotated lncRNAs, I do not think it is the best approach. As mentioned before, syntenic lncRNAs represent lncRNAs with preserved genomic positions that do not necessarily show conservation at the primary sequence level. They are actually characterised by large variations at the sequence level, including overall transcript length and the number of exons. For example, positionally conserved CASC15 lncRNA that is located in the proximity of the SOX4 gene in the human, mouse and zebrafish genomes has highly variable transcript structure in each of these species, which largely affects its length. The genomic length of CASC15 lncRNA in human, mouse and zebrafish is ~531, ~24 and ~21kb, respectively. Thus, the quality annotation assessment using syntenic lncRNAs across different species, even closely related ones can be misleading. Positional conservation also covers cis-acting lncRNAs, which can regulate neighbouring in a sequence independent way, through the act of transcription itself. Instead, I propose to simply compare the novel lncRNA genes/transcripts to the annotated ones in the macaque genome (Figure 1 and S2).

Considering the reviewer's concerns we have finally decided to exclude this analysis from the manuscript.

2. I am very grateful for including the analysis of intron length (Figure S2A). However, the comparison would be more meaningful and straightforward, if the lncRNA classes match the ones presented in the Figure 1D-E. In this way it is hard to directly compare newly annotated transcripts, their exons and introns to the annotated ones. Alternatively, distinguishing intergenic and antisense lncRNA classes among the annotated ones in Figure S2 would also help. Nevertheless, it looks like on average the novel transcripts are slightly shorter than the annotated ones, which I believe is expected

and ok. Finally, if for some reason, the authors decide to keep the transcript length comparison using syntenic lncRNAs, the Figure S2E should also include annotated macaque lncRNA orthologues.

As suggested by the reviewer, we have added the plot as Supplemental Figure 2A and kept the comparison between intergenic and antisense lncRNA as Supplemental Figure 2E.

3. Please add the absolute numbers to Figure S1C and D.

As suggested, we have added the absolute numbers in Figures S1C and S1D as shown below.

4. Please check the citation of Figure S6E, as presented numbers do not match the ones on the Figure. I think they actually refer to Figure 3B. Again, having the numbers displayed on the plot would help a lot.

We believe both the reported numbers in the text and in the figures are correct. However, we acknowledge that not adding the numbers in Figure S6E might have been confusing. Thus, we have added the numbers in Figure S6E and in all the other Supplementary figures from S6.

5. For clarity reasons, can you please label the panels in Figure S7 to show which or them refer to human and macaque?

We have labeled the panels as suggested, both in Figure S7 and Figure S4. An example of a plot is shown below.

Reviewer #2 (Remarks to the Author):

Santus et al have charted extensively lncRNAs in macaques. The publication will be a major resource for the community in the coming years. In the revised version, Santus et al have very nicely elevated my concerns regarding the discrepancy between their analysis and the publication from Sandberg and co (PMID: 35241826).

We would like to thank the reviewer for their positive evaluation of the improved work.

Minor comments:

- I would remove claims for novelty “upsilon, a novel metric”.

Following the reviewer’s suggestion we have removed novelty claims for Upsilon in the results section both in the title and text, as well as in the discussion.

- Figure 4C: does it worth to add the baseline expression too in the heatmap?

The plot in Figure 4C is colored based on log fold changes, not raw expression values. Specifically, the heatmap shows the log fold changes computed by dividing baseline expression values and expression values for each infection stage -early (E), middle (M) and late (L). We have modified the figure caption to better describe what the heatmap shows

“Heatmaps display lncRNAs DE in monocytes, T cells, and B cells in at least one infection stage -early (E), middle (M), or late (L)- as compared to baseline (B). Cells are colored according to the fold changes (log₂) in expression values between baseline and the corresponding infection stage.”

REVIEWERS' COMMENTS

Reviewer #1 (Remarks to the Author):

I support the publication of this nice and interesting manuscript in its current form.